# A point mutation in *recC* associated with subclonal replacement of carbapenem-resistant *Klebsiella pneumoniae* ST11 in China

Kai Zhou[1,60] ✉, Chun-Xu Xue [1,60], Tingting Xu [1,60], Ping Shen [2,60], Sha Wei [1], Kelly L. Wyres [3], Margaret M. C. Lam[3], Jinquan Liu [1], Haoyun Lin [4], Yunbo Chen [2], Kathryn E. Holt[3,5], the BRICS Working Group* & Yonghong Xiao[2] ✉

Adaptation to selective pressures is crucial for clinically important pathogens to establish epidemics, but the underlying evolutionary drivers remain poorly understood. The current epidemic of carbapenem-resistant *Klebsiella pneumoniae* (CRKP) poses a significant threat to public health. In this study we analyzed the genome sequences of 794 CRKP bloodstream isolates collected in 40 hospitals in China between 2014 and 2019. We uncovered a subclonal replacement in the predominant clone ST11, where the previously prevalent subclone OL101:KL47 was replaced by O2v1:KL64 over time in a stepwise manner. O2v1:KL64 carried a higher load of mobile genetic elements, and a point mutation exclusively detected in the *recC* of O2v1:KL64 significantly promotes recombination proficiency. The epidemic success of O2v1:KL64 was further associated with a hypervirulent sublineage with enhanced resistance to phagocytosis, sulfamethoxazole-trimethoprim, and tetracycline. The phenotypic alterations were linked to the overrepresentation of hypervirulence determinants and antibiotic genes conferred by the acquisition of an *rmpA*-positive pLVPK-like virulence plasmid and an IncFII-type multidrug-resistant plasmid, respectively. The dissemination of the sublineage was further promoted by more frequent inter-hospital transmission. The results collectively demonstrate that the expansion of O2v1:KL64 is correlated to a repertoire of genomic alterations convergent in a subpopulation with evolutionary advantages.

*Klebsiella pneumoniae* is a significant nosocomial pathogen worldwide, and its remarkable ability to acquire antibiotic resistance largely facilitates its widespread dissemination. In the last decade, the rate of multidrug-resistant (MDR) *K. pneumoniae*, particularly carbapenem-resistant *K. pneumoniae* (CRKP), is trending upwards globally, and is associated with an enormous global public health burden[1–3]. In particular, bloodstream infections (BSI) caused by CRKP highly challenges clinical treatments, resulting in a high mortality rate of up to over 50%

A full list of affiliations appears at the end of the paper. *A list of authors and their affiliations appears at the end of the paper.
✉ e-mail: zhouk@mail.sustech.edu.cn; xiaoyonghong@zju.edu.cn

in nosocomial settings[4,5]. The World Health Organization has included CRKP in a list of antimicrobial-resistant priority pathogens for which new antibiotics are urgently needed.

The rapid expansion of CRKP has been attributed to the acquisition of carbapenemases as well as the establishment of successful clones (i.e., high-risk clones). The population structure of CRKP varies geographically[6]. In Asia, especially China, KPC-2-producing sequence type (ST) 11 is predominant, accounting for up to 60–70% of CRKP[3]. ST258, a supposed descendent of ST11, has become the most prevalent KPC-2/KPC-3 producing clone in North America, Latin America, and Europe[2,6,7]. Although these clones have remained at high prevalence in certain regions for decades, intra-clonal segregations have been observed. More than two subclones have been identified in the ST11 and ST258 population, and recombinations involving the capsule polysaccharide synthesis (CPS) locus are supposed to be primarily responsible for genetic diversification[4,8,9]. We recently revealed a subclonal switch among CRKP-ST11 bloodstream isolates collected in a single center in China between 2013 and 2017, where ST11-KL47 had been displaced by ST11-KL64 as the predominant subclone[4]. Of greater concern, ST11-KL64 has evolved enhanced pathogenicity, resulting in significantly higher 30-day mortality compared to ST11-KL47. However, the spatiotemporal dynamics and underlying driving forces of the population structure remain poorly understood.

In this work, we investigate the genomic evolution of 794 CRKP isolates collected in the framework of national surveillance for bloodstream isolates between 2014 and 2019 across China to elucidate the spatial and temporal dynamic population structure of CRKP-ST11 and to dissect the genetic and phenotypic drivers of the intra-clonal diversification. The genomic alterations correlating with subclonal switch and phenotypical variations in the dominant ST11 population are characterized and linked to evolutionary drivers.

## Results

### Population structure of CRKP bloodstream (CRKP-BS) isolates in China

Between 2014 and 2019, 4635 *K. pneumoniae* species complex bloodstream isolates were collected from 45 sentinel hospitals distributed across 19 provinces covering 75.7% population of China (ca. 1.06 billion) (Fig. 1a). A total of 794 non-repetitive CRKP isolates were identified in 40 hospitals of 16 provinces, including 772 *K. pneumoniae* sensu stricto, 10 *K. variicola*, 11 *K. quasipneumoniae,* and 1 *K. michiganensis* (not belonging to *K. pneumoniae* species complex but was included in the analysis) (Supplementary Fig. 1 and Supplementary Dataset 1). The proportion of CRKP had increased from 11.2% to 17.7% over the study period (Fig. 1b). The population structure of CRKP-BS was highly complex, and 72 STs were detected (Supplementary Dataset 1). ST11 was the predominant clone (81.4%; 646/794), followed by ST15 (55/794; 6.93%).

One or more carbapenemase genes were detected in 771 of 794 isolates (97.1%), and 753 belonged to *K. pneumoniae* sensu stricto, which encoded $bla_{KPC}$-like ($n = 712$; including 709 $bla_{KPC-2}$ and 3 $bla_{KPC-3}$), $bla_{NDM}$-like ($n = 36$), $bla_{IMP}$-like ($n = 4$), and $bla_{OXA-48}$-like ($n = 5$) genes (Supplementary Dataset 1). The majority of $bla_{KPC}$-like-positive *K. pneumoniae* sensu stricto isolates (686/712; 96.3%) belong to ST11 and ST15, suggesting clonal dissemination of $bla_{KPC}$-like genes in China.

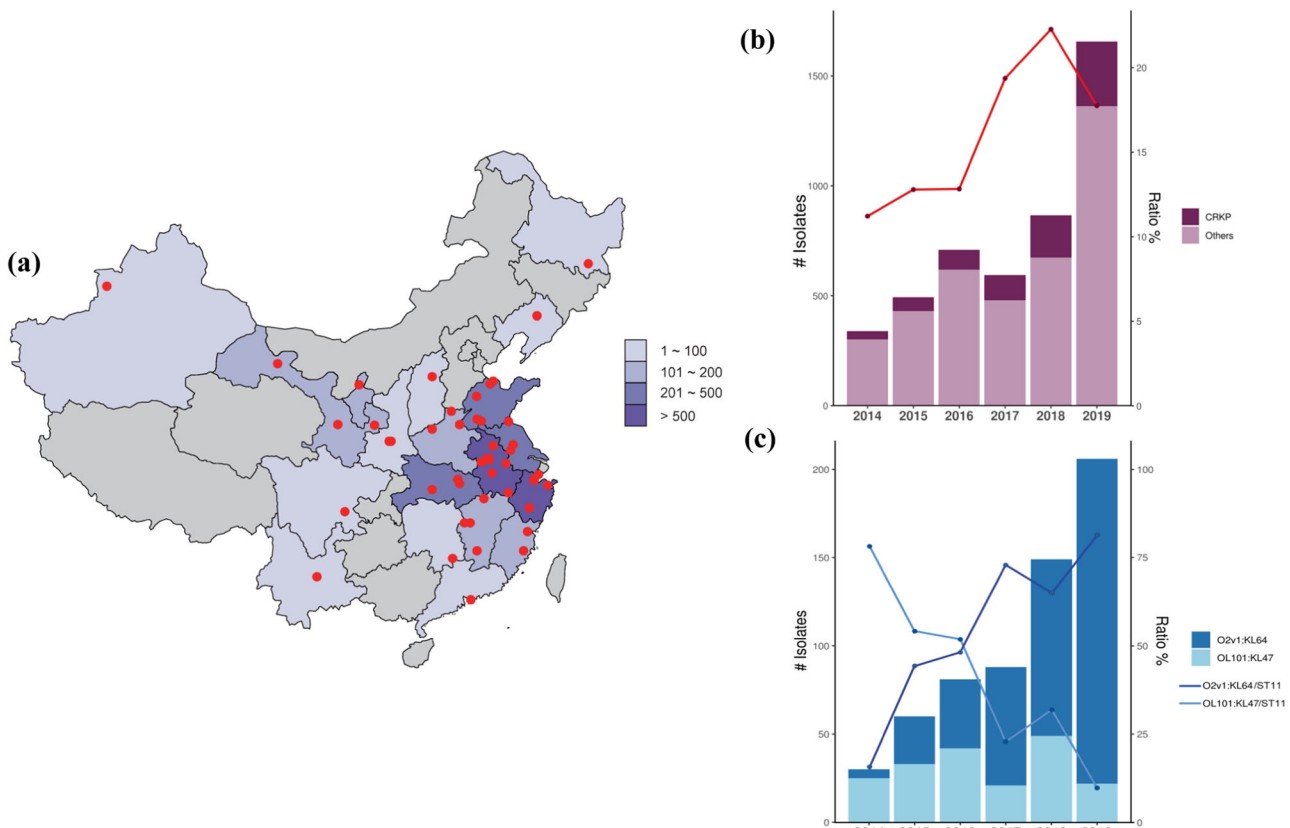

**Fig. 1 | *K. pneumoniae* species complex bloodstream isolates collected in this study between 2014 and 2019. a** Geographical distribution of 45 sentinel hospitals participating in this study. The number of isolates collected in each province is shown by color gradients at the right. **b** The graph shows the number (dark red bars) and the ratio of CRKP (red line) detected each year during the surveillance. **c** The graph shows the number of O2v1:KL64 and OL101:KL47 detected in CRKP (blue bars) each year, and the ratio of O2v1:KL64 to CRKP-ST11 (purple line) and OL101:KL47 to CRKP-ST11 (blue line). Source data are provided as a Source Data file.

## Occurrence of subclonal switch in CRKP-ST11 within a 6-year period

We identified 55 K-loci (KLs, capsule synthesis loci) and ten O-loci (OLs, outer lipopolysaccharide synthesis loci) in the CRKP-BS population (Supplementary Dataset 1). ST11 comprised 13 KLs and 8 OLs, of which KL64 (422/646; 65.3%) and O2v1 (418/646; 64.8%) was the most prevalent, followed by KL47 (192/646; 29.7%) and OL101 (an O12 derivative) (193/646; 29.9%). ST15 included 5 KLs and 5 OLs, and KL19 (43/56; 76.8%) and O2v1 (42/56; 75%) were predominant. The proportion of OL101:KL47 among CRKP-ST11 dropped from 78.1% (25/32) in 2014 to 9.7% (22/226) in 2019, whereas that of O2v1:KL64 (4 OL102:KL64 isolates were included to simplify the analysis through the study) increased from 15.6% (5/32) in 2014 to 81.4% (184/226) in 2019 (Fig. 1c). The findings demonstrate that subclonal replacement has occurred from OL101:KL47 to O2v1:KL64 within the ST11 population in China. The subclone O2v1:KL19 was constantly prevalent in ST15 during the study period (66.7%-87%).

## O2v1:KL64 is derived from OL101:KL47

To determine the phylogenetic relationship of these ST11 subclones, a maximum-likelihood tree was derived from 4460 recombination-free SNPs. A clade comprising isolates of O2v1:KL103, O2v2:KL105, O3b:KL111, and O4:KL15 was basal in the tree (Fig. 2), presenting the ancestral clade of the ST11 isolates. All O2v1:KL64 isolates clustered together to form the deepest branching clade and also clustered with one sublineage of OL101:KL47, indicating that O2v1:KL64 was derived from OL101:KL47. This is consistent with our previous conclusion from single-center data[4]. The other serotypes clustered either with OL101:KL47 (O3/O3a:KL10 and O5:KL25) or with O2v1:KL64 (O2v1:KL21, O2v1:KL28, O2v1:KL31, O2v1:KL103, O2v1:KL107, and O3/O3a:KL58), supporting the notion that they evolved from the two major subclones.

## Recombination contributes significantly to the intra-clonal diversification of CRKP-ST11

We identified 42,824 core-genome SNPs prior to and 4460 SNPs after the removal of recombination regions, suggesting that recombination has contributed heavily to the population diversity. These include a 96.1-kb and 12.1-kb region encompassing the CPS and lipopolysaccharide (LPS) locus that introduced 1182 and 261 SNPs, respectively, thereby accounting for the switch from OL101:KL47 to O2v1:KL64 (Supplementary Fig. 2). There were additional recombination events spanning the CPS and/or LPS region that were detected in the OL101:KL47/O2v1:KL64-derived subclones and these also conferred switches in O/K-types.

To further estimate the role of recombination in the genetic variations of O2v1:KL64 and OL101:KL47, we calculated nucleotide divergence for all pairs of genomes within the two subclones before and after the removal of recombinant sequence regions. Less nucleotide divergence was detected in O2v1:KL64 than in OL101:KL47 before (median pairwise divergence: $4.2 \times 10^{-5}$ vs $1.5 \times 10^{-4}$; $p < 2.2 \times 10^{-16}$ by Wilcoxon rank-sum test) and after the removal of recombination regions ($1.3 \times 10^{-5}$ vs $2.1 \times 10^{-5}$; $p < 2.2 \times 10^{-16}$) (Supplementary Fig. 3). This is probably due to the fact that O2v1:KL64 emerged later than OL101:KL47 and has had less time to accumulate genetic diversity. The r/m value of O2v1:KL64 was approximately 2.5-fold higher than that of OL101:KL47 (17.68 vs 7.28), suggesting that the contribution of recombination to the genetic variations was higher in O2v1:KL64 than in OL101:KL47.

## Point mutation in RecC confers a higher recombination frequency to O2v1:KL64

Given the higher r/m value detected in O2v1:KL64 compared with OL101:KL47, and more serotypes derived from O2v1:KL64 ($n = 7$) than from OL101:KL47 ($n = 2$), we supposed that O2v1:KL64 might have

evolved with higher recombination proficiency. To test the hypothesis, we examined subclone-specific SNPs associated with recombination, and a single missense mutation in the *recC* gene (2804 A > G; His935Arg) was exclusively found in O2v1:KL64 compared with the sequence of OL101:KL47. It is known that functional *recC* is required for genetic recombination in *Escherichia coli*, and mutations in *recC* can affect recombination proficiency[10]. We engineered an O2v1:KL64 mutant (KP37485Δ*recC*), and confirmed that deletion of *recC* indeed abolished recombination proficiency reflected by the resistance to the DNA-damaging agent mitomycin C[11], which could be restored by complementation with $recC_{O2v1:KL64}$ or $recC_{OL101:KL47}$ (Fig. 3), demonstrating that *recC* is involved in recombination in *K. pneumoniae* as that in *E. coli*.

To validate whether the single missense mutation of *recC* could affect recombination proficiency, we engineered an isogenic mutant KP37485-$recC_{OL101:KL47}$ by replacing the $recC_{O2v1:KL64}$ with $recC_{OL101:KL47}$. Apart from 2804 A > G, no other SNPs were found in the genome of KP37485-$recC_{OL101:KL47}$ confirmed by sequencing. Compared with that of KP37485, a threefold reduction of resistance to mitomycin C was observed for KP37485-$recC_{OL101:KL47}$ [survival ratio: $(1.85 \pm 0.38) \times 10^{-6}$ vs $(0.62 \pm 0.05) \times 10^{-6}$); $p = 0.0295$ by $t$-test], indicating that the single mutation in *recC* gene has an effect on recombination proficiency. We further designed a transduction experiment to validate the impact of the two *recC* alleles on recombination frequency (Supplementary Fig. 4). KP37485 showed a 245-fold higher recombination frequency than KP37485-$recC_{OL101:KL47}$ [mean $(1.38 \pm 0.48) \times 10^{-8}$ vs $(5.62 \pm 0.44) \times 10^{-11}$; $p = 0.0388$ by $t$-test], while homologous recombination was undetectable in KP37485Δ*recC*. The results demonstrate that the single missense mutation in the *recC* gene can significantly enhance recombination proficiency.

We further analyzed 14,407 *K. pneumoniae* genomes retrieved from the NCBI RefSeq database as of November 2022 to determine the distribution of $recC_{His935Arg}$, and the allele was exclusively found in 763 of 823 ST11 O2v1:KL64 isolates (Supplementary Dataset 2). All but one of the $recC_{His935Arg}$-positive isolates were collected in China, indicating that $recC_{His935Arg}$ was specific to Chinese ST11 O2v1:KL64 isolates.

## O2v1:KL64 encodes a higher load of mobile genetic elements (MGEs)

Recombination and horizontal gene transfer are known to be vital in shaping bacteria genome structures by affecting the exchange of genetic materials, e.g., MGEs[12]. We here analyzed MGEs in the 646 ST11 genomes, including integrons, insertion sequences (ISs), prophages, and plasmids (approximated by replicons), to identify additional mechanisms involved in the diversification of CRKP-ST11. A significantly higher load of prophages, replicons, and ISs was found in O2v1:KL64 compared to OL101:KL47 (median 9 vs 8; 6 vs 4; and 22 vs 18, respectively) (Wilcoxon rank-sum test: $p = 2.2 \times 10^{-16}$; $2.2 \times 10^{-16}$; $1.87 \times 10^{-8}$) (Supplementary Fig. 5a–d), while the number of integrons was comparable in the two subclones with a significantly different distribution (median 2 vs 2; $p = 6.88 \times 10^{-9}$) (Supplementary Fig. 5e). A similar trend was found between ST11 and no-ST11 (Supplementary Fig. 6). To better understand whether the differences in prophages, plasmids, and ISs between the two subclones were due to vertical or horizontal transfer, we reconstructed the ancestral states of these MGEs. Our results indicate that the large-scale expansion of plasmids and ISs within O2v1:KL64 was likely caused by a single acquisition event (Fig. 4a, b), supporting the vertical model. In contrast, we found that differences in prophages were likely due to multiple acquisition and loss events within O2v1:KL64 (Fig. 4c), suggesting a more complex pattern of horizontal transfer. These findings support the critical role of MGEs, especially of prophages and plasmids, in shaping the population structure of CRKP-ST11.

It is known that phage-induced selective pressures play a critical role in driving the serotype switch of *K. pneomoniae*[13]. We, therefore,

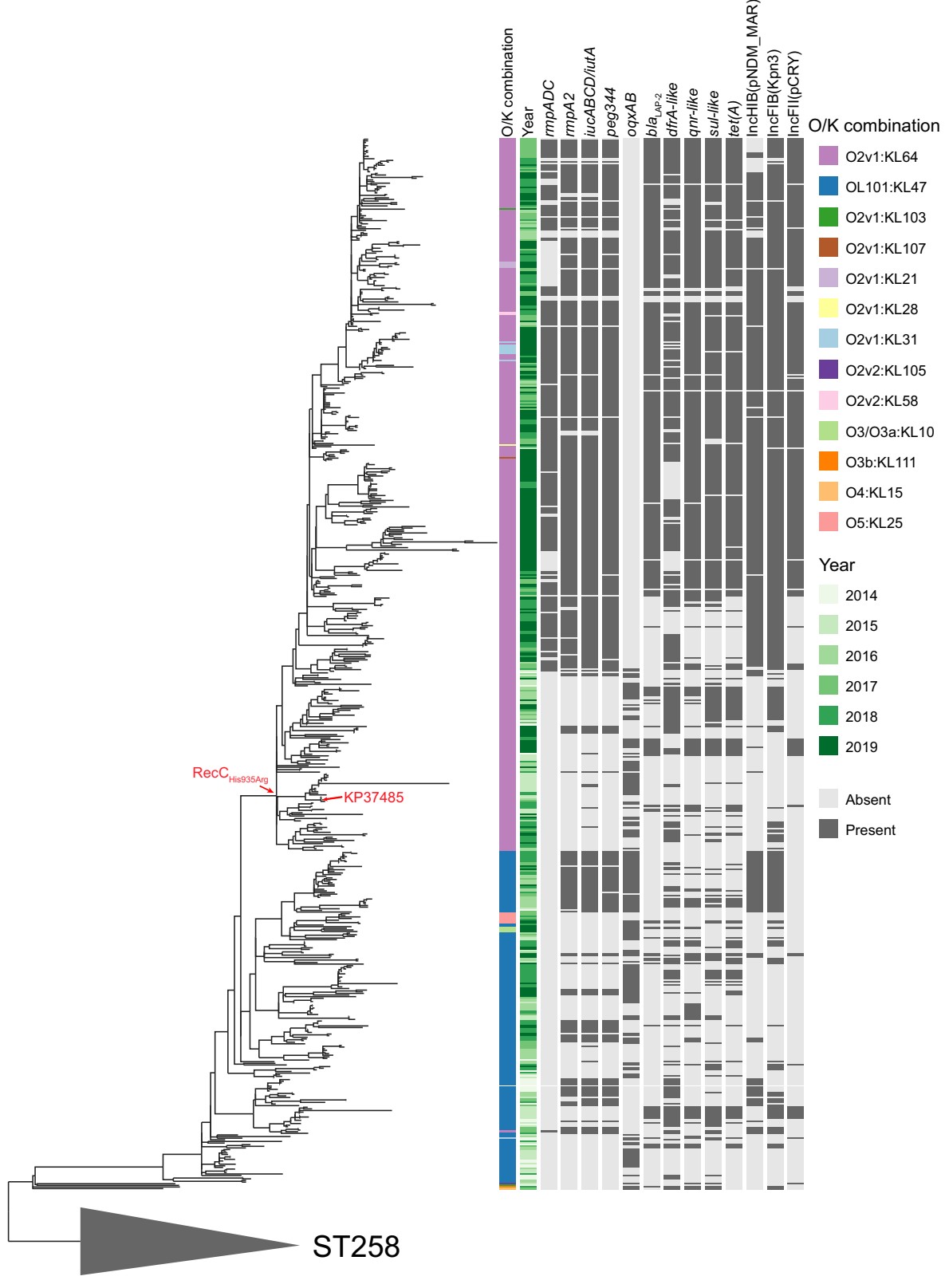

**Fig. 2 | Phylogenetic analysis of 646 CRKP-ST11 isolates collected in this study.** The phylogenetic tree was obtained by mapping all sequence reads to the hybrid assembly of an ST11-OL101:KL47 isolate (KP16932) and removing the recombined regions from the alignment. The tree was rooted using the ST258 outgroup isolates (gray triangle). Thirteen O/K combinations were detected in our ST11 collection, which are indicated in different colors, as shown in the legend. The hypervirulence biomarkers detected (except for *iro* due to its rarity in our collection) are shown here. The ARGs [*bla*<sub>LAP-2</sub>, *dfrA*-like, *qnr*-like, *sul*-like, *tet(A)*, and *oqxAB*] detected with

significantly different abundance between OL101:K47 and O2v1:KL64 are shown. The replicon types corresponding to the prevalent virulence [IncHI1B(pNDM-MAR) and IncFIB(Kpn3)] and MDR plasmid [IncFII(pCRY)] carrying these ARGs (except for *oqxAB*) are shown here. The RecC allele (RecC<sub>His935Arg</sub>) was detected exclusively in O2v1:KL64, as shown on the branch, and the reference (KP37485) used in the recombination assay is indicated on the tree. Source data are provided as a Source Data file.

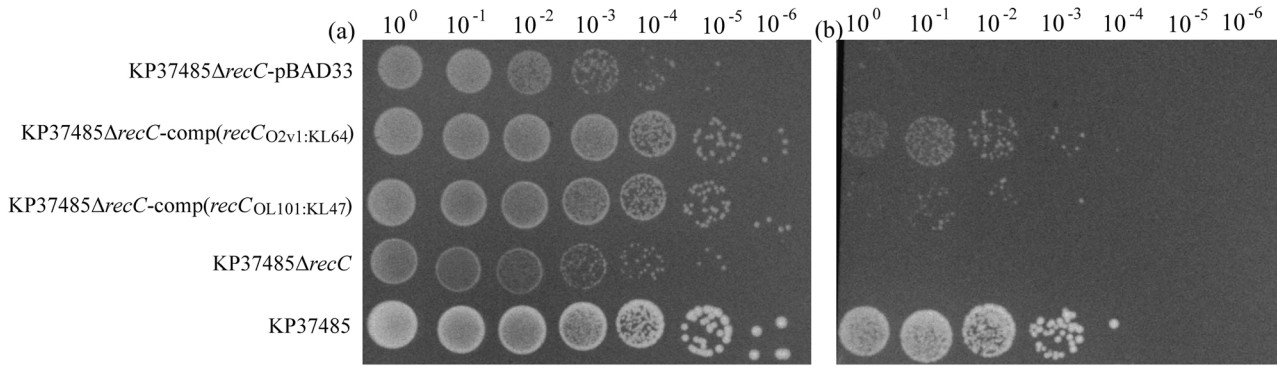

**Fig. 3 | Mitomycin C resistance mediated by *recC*.** Late logarithmic phase cells grown in LB broth with 0.2% (w/v) L-arabinose and 25 mg/L chloramphenicol were harvested and resuspended in PBS to obtain $5 \times 10^8$ cfu/ml, followed by being treated with PBS (**a**) or mitomycin C at 8 mg/L (**b**) for 1 h. Cultures were serially diluted tenfold, spotted at 10 ul in rows on LB plates, and incubated overnight at 37 °C. Source data are provided as a Source Data file.

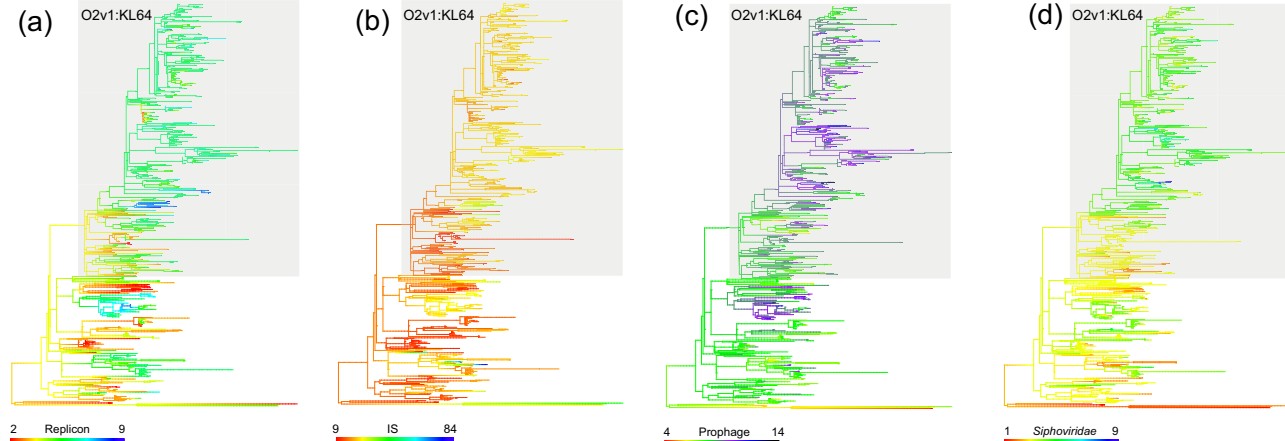

**Fig. 4 | Ancestral state reconstructions of MGEs with significant differences between O2v1:KL64 and OL101:KL47.** MGE was mapped as a continuous character onto the phylogenetic tree calculated by the 646 ST11 genomes. Evolution reconstructed with the R package phytools on the dataset, including the number of plasmid replicons (**a**), IS copies (**b**), prophages (**c**), and *Siphoviridae* prophages (**d**) identified per genome. Colors are assigned based on the number of MGEs detected per genome, as indicated by the bars. The O2v1:KL64 subclone is highlighted by a gray box. Source data are provided as a Source Data file.

examined whether the presence and type of phages were possibly involved in the intra-clonal diversification of CRKP-ST11. Most of the prophages detected (95.03%) were classified into three families, namely *Myoviridae*, *Podoviridae*, and *Siphoviridae*. A load of *Myoviridae* and *Podoviridae* prophages was comparable between the two subclones (median 92 kb; 19.8 kb), while that of *Siphoviridae* prophages was significantly higher in O2v1:KL64 than in OL101:KL47 (median 95 kb vs 60.3 kb) ($p = 2.2 \times 10^{-16}$ by Wilcoxon rank-sum test) (Supplementary Fig. 5f–h). As with plasmids and ISs, the widespread expansion of *Siphoviridae* prophages probably resulted from a single acquisition event within O2v1:KL64 (Fig. 4d).

**Emergence of a hypervirulent population by exclusively obtaining *rmpA*-positive virulence plasmids drives the expansion of O2v1:KL64**

To evaluate the potential pathogenesis of O2v1:KL64 and OL101:KL47, 154 experimentally validated virulence factors (VFs) of *K. pneumoniae* were analyzed (see methods). O2v1:KL64 carried significantly more virulence determinants than OL101:KL47 (median 97 vs 85) ($p = 2.2 \times 10^{-16}$ by Wilcoxon rank-sum test) (Supplementary Fig. 7). A set of key VFs (i.e., *iucABCD*, *terABCDEZ*, *peg-344*, *rmpADC*, and *rmpA2*) associated with hypervirulence and typically mobilized by virulence plasmids[8,14] mainly contributed to the inter-subclonal differences (Supplementary Dataset 3), and their proportions were significantly

higher in O2v1:KL64 (56.9–71.1%) than in OL101:KL47 (0–41.1%) ($p \leq 2.82 \times 10^{-9}$ by Chi-square test for each pairwise comparison). Notably, *rmpADC* was exclusively carried by O2v1:KL64 with a rate of 56.9% (240/422). Frame-shifted *rmpA* and incomplete *rmpADC* were detected in 21.3% (51/240) and 9.2% (22/240) *rmpADC*-positive O2v1:KL64 isolates, respectively. While *rmpA2* was found in OL101:KL47 (74/192) and O2v1:KL64 (290/422), frame-shifted *rmpA2* was detected in 67.6 and 98.6% of *rmpA2*-positive OL101:KL47 (50/74) and O2v1:KL64 (286/290) isolates, respectively. This is similar to our previous observation from single-center data[4]. Since *rmpA/A2* is frequently used as the indicator of virulence plasmids, we here simply defined the *rmpA/A2*-positive isolates as hvKP. The O2v1:KL64-hvKP isolates emerged in 2015 and were detected in nine provinces. The proportion of O2v1:KL64-hvKP among O2v1:KL64 and CRKP-ST11 dramatically increased from 0% in 2014 to 85.9% (158/184) and 69.9% (158/226) in 2019 (Supplementary Fig. 8), respectively, suggesting that the expansion of O2v1:KL64 was associated with an increase in the size of the hvKP population.

To confirm the existence of virulence plasmids in ST11, reads of the 646 ST11 genomes were mapped to two representative virulence plasmids pVir-KP16932 (carried by an ST11-OL101:KL47 isolate) and pVir-KP47434 (carried by an ST11 O2v1:KL64 isolate) reported in our previous study[4], and the presence of a virulence plasmid was inferred if the coverage was ≥40% of the reference. We identified 400 isolates

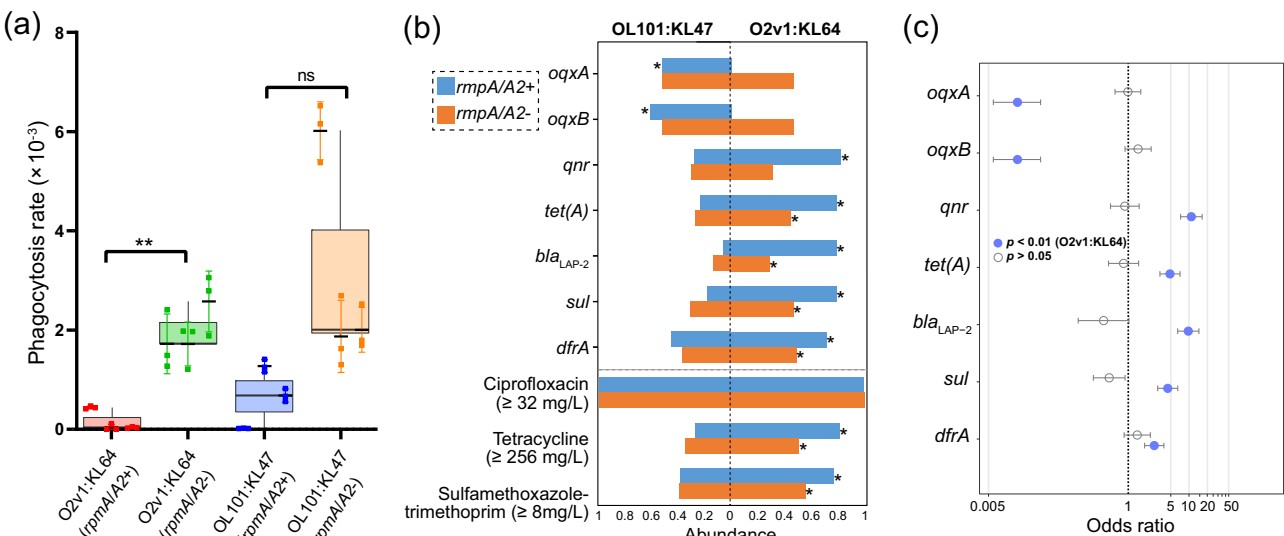

**Fig. 5 | Phenotypes and genotypes with significant differences between OL101:KL47 and O2v1:KL64. a** Comparison of phagocytosis resistance for isolates w/o virulence plasmids (indicated by the presence of *rmpA/A2*). Three isolates were randomly selected for each group in the test, i.e., O2v1:KL64-*rmpA/A2*+ (KP33367, KP47434, and KP66639), O2v1:KL64-*rmpA/A2*- (KP33316, KP37485, and KP39199), OL101:KL47-*rmpA/A2*+ (KP16932, KP41051, and 46882), and OL101:KL47-*rmpA/A2*- (KP30412, KP43350, and KP73269). The assay was triplicated, and the error bars represent standard deviations. Student's *t*-tests were used for pairwise group comparisons, and *rmpA/A2*-positive O2v1:KL64 isolates showed significantly enhanced resistance to phagocytosis than *rmpA/A2*-negative O2v1:KL64 isolates ($p = 0.0043$). Boxplots are displayed using the Tukey method (center line, median; box limits, upper and lower quartiles; whiskers, last point within a 1.5x interquartile range). ns, not significant; *$p < 0.01$; **b** Distribution of ARGs with significant differences between OL101:KL47 and O2v1:KL64 w/o virulence plasmids. The resistance ratio of drugs (ciprofloxacin, tetracycline, and sulfamethoxazole-trimethoprim) associated with these ARGs are compared. Since the breakpoint of tetracycline is not available for *K. pneumoniae*, we here used MIC ≥ 256 mg/L to represent high-level resistance. *$p < 0.01$ (Chi-squared test). **c** ARGs associated with virulence plasmids (indicated by the presence of *rmpA/A2*) in OL101:KL47 ($n = 192$) and O2v1:KL64 ($n = 422$). Circles indicate odds ratios estimated in a single multivariable logistic regression model with all genes; lines indicate 95% confidence intervals for those odds ratios. The median was used as the measure of center for the error bars in panels **a** and **c**. All statistical tests carried out were two-sided. Source data are provided as a Source Data file.

(61.9%) which might carry a virulence plasmid (Supplementary Dataset 4). Of these, 33 isolates (13 OL101:KL47 and 20 O2v1:KL64) with mapping coverage ranging between 40–100% were randomly selected for long-read sequencing to examine the diversity of virulence plasmids (Supplementary Dataset 5). The *rmpA/A2* genes were detected on the chromosome of 3 OL101:KL47 isolates, and on a putative virulence plasmid in each of 30 isolates, ranging in size from 101.5 to 305.5 kb (Supplementary Dataset 6). The 30 putative virulence plasmids were typed as IncFIB-HIB ($n = 25$), IncFIB ($n = 2$), IncFII-FIB-R ($n = 2$), and untypeable ($n = 1$), and were grouped into four clusters by MOB-suite[15] with AA406 as the predominant cluster (Supplementary Dataset 6). Of these, 13 plasmids shared a relatively conserved backbone (75.5–100% coverage), another 10 with a smaller size could be derived from them or vice versa by gain or loss of genes (67.3–100% coverage), and the other seven were resistance-virulence fusion plasmids carrying 1–10 antimicrobial resistance genes (ARGs) with a lower coverage to the virulence plasmid references (≤60%) (Supplementary Dataset 6 and Supplementary Fig. 9). Mapping the short sequence reads of 389 *rmpA/A2*-positive ST11 isolates to the 30 putative virulence plasmid sequences identified 347 (89.2%) showing ≥90% coverage to a circularized 107.1-kb IncFIB plasmid pVir-KP115906 which yielded the highest number of matches. These data indicate that most *rmpA/A2*-positive isolates harbored virulence plasmids. Comparing genome phylogenetic positions and plasmid sequence similarities indicated both horizontal and vertical modes of virulence plasmid transmission among OL101:KL47 and O2v1:KL64 (Supplementary Fig. 9).

### Virulence plasmids are associated with significantly enhanced resistance to phagocytosis in O2v1:KL64

It is known that virulence plasmids can promote the pathogenicity of *K. pneumoniae*; we here measured phagocytosis to evaluate the pathogenicity of isolates w/o virulence plasmids. Isolates of each subclone

were grouped by the presence of *rmpA/A2* (the indicator of virulence plasmids), and three isolates were randomly selected for each group in the test. Indeed, compared with those without virulence plasmids, *rmpA/A2*-positive isolates of both subclones showed enhanced resistance to phagocytosis, but it was only significant for O2v1:KL64 (mean phagocytosis 0.017 ± 0.023% vs 0.2 ± 0.049%; $p = 0.0043$ by two-sided *t*-test) (Fig. 5a).

### Tetracycline and sulfamethoxazole-trimethoprim may have been involved in the selection of O2v1:KL64-hvKP

To explore whether antibiotics were involved in the subclonal selection, we analyzed acquired antibiotic resistance genes (ARGs) for both subclones. The number of acquired ARGs was comparable between O2v1:KL64 (median 11) and OL101:K47 (median 10) ($p = 0.96$ by Wilcoxon rank-sum test) (Supplementary Fig. 10). However, of these with ≥50% proportion in either subclone, the abundance of *oqxAB* was significantly higher in OL101:KL47 ($p \leq 5.69 \times 10^{-12}$ by Chi-square test), while that of *bla*$_{LAP-2}$, *dfrA*-like, *qnr*-like, *sul*-like, and *tet(A)* was significantly higher in O2v1:KL64 ($p \leq 1.23 \times 10^{-3}$ by Chi-square test) (Fig. 5b and Supplementary Dataset 7). A strong correlation was found between the presence of these ARGs and *rmpA/A2* in O2v1:KL64 ($p < 2.2 \times 10^{-16}$), but not in OL101:KL47 ($p \geq 0.05$) (Fig. 5c). We further measured MICs of ciprofloxacin, tetracycline, and sulfamethoxazole-trimethoprim for 368 O2v1:KL64 and 140 OL101:K47 isolates. No significant differences were found for ciprofloxacin resistance between O2v1:KL64 and OL101:K47 ($p \geq 0.95$ by Chi-square test) (Fig. 5b), since all isolates harbored a *gyrA* mutant (Asp87Gly) and a *parC* mutant (Ser80Ile and Asn438Ser). However, the O2v1:KL64-hvKP population displayed the highest resistance rate to tetracycline and sulfamethoxazole-trimethoprim (Fig. 5b).

To understand how these ARGs were captured, we performed long-read sequencing for 17 isolates (14 O2v1:KL64; 3 OL101:KL47)

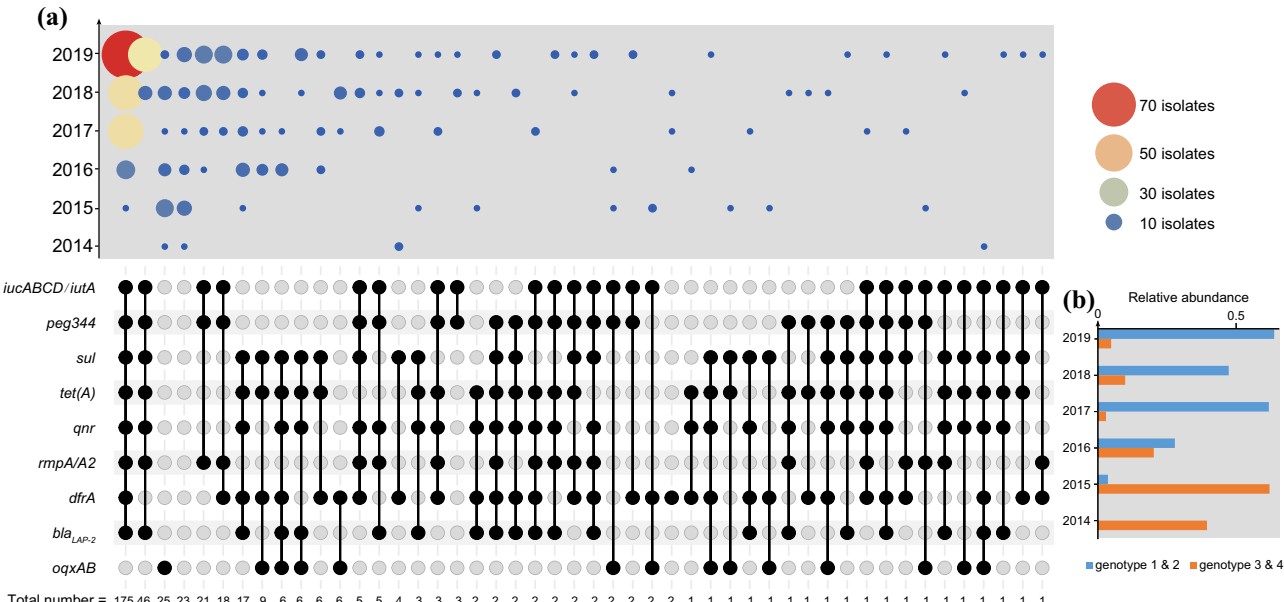

**Fig. 6 | Temporal genotype trends in O2v1:KL64 from 2014 to 2019. a** The combination matrix (bottom) depicts predicted genotypes of O2v1:KL64. Within the combination matrix, black circles indicate the presence of genes and the vertical combination of black circles represents the genotype. The total number of each genotype is indicated below the matrix. The combination matrix was created using the UpSetR package[73]. The bubble plot (top) depicts the relative number of isolates with each genotype (combination matrix) per year. **b** The graph shows the relative abundance of genotypes 1 and 2 and genotypes 3 and 4 per year. Source data are provided as a Source Data file.

carrying at least two of $bla_{LAP-2}$, $dfrA$-like, $qnr$-like, $sul$-like, and $tet(A)$ genes. These ARGs were detected on the replicon-encoding contigs of the 17 genomes, supporting that they were plasmid-borne (Supplementary Dataset 8). These putative plasmids were assigned to three Inc types (IncFII, IncFII-FIB, and IncFII-R) and IncFII was predominant (13/18). Each type of plasmid shared a conserved backbone irrespective of hosts, suggestive of inter-subclonal horizontal transfers (Supplementary Fig. 11a). Mapping the reads of OL101:KL47 and O2v1:KL64 genomes to these putative plasmids revealed that the IncFII-type plasmid was prevalent in O2v1:KL64 (266 genomes showed >90% coverage to pMDR-KP29007), especially in O2v1:KL64-hvKP (241/266), but rare in OL101:KL47 (12 genomes showed >90% coverage to pMDR-KP29007) (Supplementary Fig. 11b).

### Detection of successful genotypes in O2v1:KL64
Given that different genotypes were conferred by the genetic diversity of the IncFIB-type virulence plasmids and IncFII-type MDR plasmids in O2v1:KL64 described above, we intended to identify successful genotypes in the context of the associated genes carried by these plasmids (Fig. 6). A total of 48 genotypes was detected based on various combinations of these genes. The most prevalent genotypes (genotypes 1 and 2) encode all genes but $dfrA$-like and/or $oqxAB$ accounting for 47.9% (202/422) of O2v1:KL64 (Fig. 6a), and the ratio of the two genotypes in each year dramatically increased from 0% to 64.7% (119/184) between 2014 and 2019 (Fig. 6b), suggesting that both could be successful genotypes. In contrast, the third and fourth prevalent genotypes (genotypes 3 and 4) do not carry any target genes or merely encode $oqxAB$ (Fig. 6a), and their proportion in the population decreased from the peak (63%; 17/27) in 2015 to 4.9% (9/184) in 2019 (Fig. 6b).

### Enhanced inter-hospital transmission promotes O2v1:KL64-hvKP dissemination
To understand whether the subclonal replacement is associated with an altered transmission pattern, i.e., intra- and inter-hospital transmission, we tried to discriminate likely recent transmission events using pairwise SNP distances by year (2014–2019). We tested a range of SNP thresholds to minimize the bias possibly introduced by a single cutoff. The minimum SNP threshold was set to be 14 SNPs based on the mutation rate of our collection and the reference genome length (see methods), and the maximum was defined to be 25 SNPs based on recent regional epidemiologic studies[7,16–18].

A significant transmission pattern shift was observed for O2v1:KL64 from 2015 to 2019 (2014 was excluded in the analysis due to the limited number of isolates obtained) using each threshold of 14–25 SNPs. The fraction of recent transmission within hospitals dramatically decreased from (100–89.7%) to (71.5–40.9%) for O2v1:KL64, while that between hospitals dramatically increased from (0–10.3%) to (28.5–59.1%) ($p < 0.05$ by Mann–Kendall test) (Fig. 7). Similar transmission dynamics were observed for the O2v1:KL64-hvKP population ($z = 2.02$; $p < 0.05$ by Mann–Kendall test), but not for the $rmpA/A2$-negative isolates (Fig. 7). This suggests that the hypervirulent subpopulation promoted the dissemination of O2v1:KL64 through enhanced inter-hospital transmissions. Relative stability was observed in the fraction of recent transmission within ([100–98.6%]–[100–83.3%]) and between hospitals ([0–1.4%]–[0–16.7%]) for OL101:K47 from 2014 to 2019 ($p > 0.05$ by Mann–Kendall test) (Fig. 7), suggesting that the transmission pattern of OL101:K47 had no significant changes over time.

### Geographical epidemiology of ST11 subclones
To dissect the geographical epidemiology and evolutionary relationship of the ST11 at the subclonal level, phylogenetic analysis was performed using the 646 ST11 genomes together with 329 publicly available draft genomes of ST11 isolated in 43 countries across four continents (i.e., America, Africa, Asia, and Europe) (Supplementary Dataset 9). All but two Chinese isolates fall into a single clade, with seven isolates obtained from France (supposedly imported from China[19]), Canada, the United States, and Japan (Supplementary Fig. 12), suggesting that the ST11 clone evolved independently and expanded locally in China. Three O3b:KL13 isolates from the United States clustered with the Chinese clade.

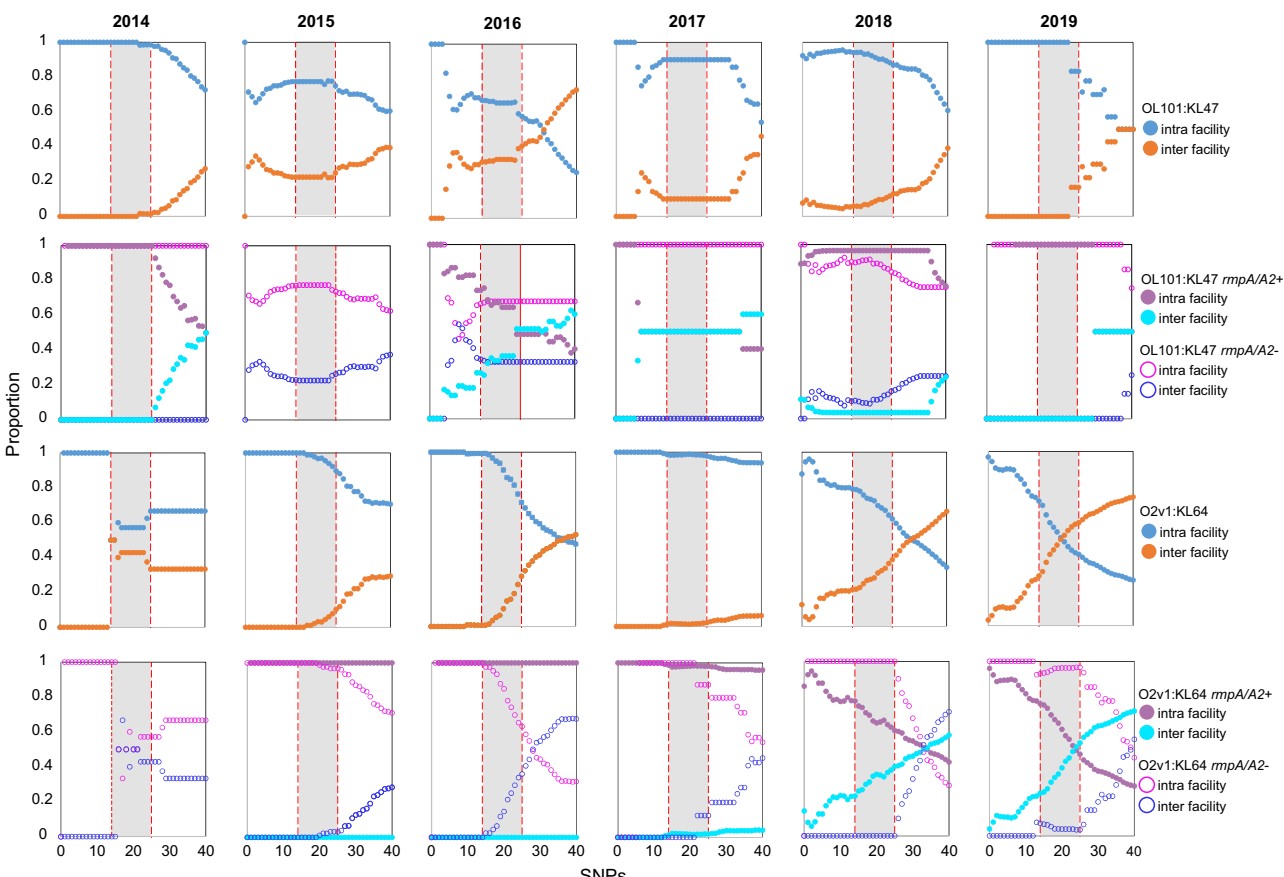

**Fig. 7 | Temporal transmission dynamics of OL101:KL47 and O2v1:KL64 (2014–2019).** Recent intra- and inter-facility transmission pairs using pairwise single-nucleotide polymorphism (SNP) distances were identified by year. Circles indicate the proportion of pairs (*y*-axis) calculated by using various pairwise SNP distance thresholds (*x*-axis). A range of thresholds (14–25 SNPs; a gray area) were used in this study (see methods) to identify recent intra- and inter-facility transmission pairs for OL101:KL47 (the top two rows) or O2v1:KL64 (the bottom two rows). Source data are provided as a Source Data file.

Most of the global ST11 isolates (269/329) were assigned to six major sublineages (Supplementary Fig. 12), including O2v1:KL24 (*n* = 66), O2v1:KL64 (*n* = 21), O2v2:KL27 (*n* = 36), O2v2:KL105 isolates (*n* = 73), O3:KL125 (*n* = 17) and O4:KL15 (*n* = 56). These sublineages displayed geographic specificity: (i) O4:KL15, O2v1:K24, and O3b:KL125 as international sublineages have spread in multiple countries across ≥3 continents (*n* = 18 across five continents; *n* = 19 across four continents; *n* = 4 across three continents); (ii) O2v2:KL105 mainly circulated in Europe, especially Eastern Europe (59/73), and O2v2:KL27 isolates were mainly detected in North and South America, suggestive of two continental; (iii) O2v1:KL64 was primarily found in Brazil (18/21) as a local sublineage. Of note, the Brazil O2v1:KL64 isolates were phylogenetically distinct from that of China, indicating an independent evolution for this subclone.

BEAST analysis was performed using 147 of the 975 genomes to reduce computation time (see Methods) (Supplementary Fig. 13). The correlation between root-to-tip distances and sampling time indicated a relatively clocklike pattern of molecular evolution (Supplementary Fig. 14). O2v1:KL64 was derived from OL101:KL47 around AD 2006 (95% HPD AD 2004–2009) in China, and OL101:KL47 emerged around AD 2005 (95% HPD AD 2002–2008). The most recent common ancestor (MRCA) of the global and Chinese isolates dated to about AD 1983 (95% HPD AD 1977–1989).

## Discussion

The rapid expansion of CRKP globally is largely driven by a number of "highly successful" clones, including ST11. However, the drivers underpinning the successful epidemic spread remain poorly understood. Tracing the genetic and phenotypic variations of isolates spanning a wide range of time and geographies allows us to investigate the evolutionary trajectory and explore the underlying driving forces.

We previously detected a subclonal shift in CRKP-ST11 causing bloodstream infections in a tertiary hospital[4]. In this study, we further demonstrated the subclonal shift over a 6-year national prevalence survey. CRKP-ST11 has diversified into two major subclones (OL101:KL47 and O2v1:KL64), and OL101:KL47 was replaced in a stepwise manner by O2v1:KL64 over time. The intra-clonal segregation would occur around AD 2006 (Supplementary Fig. 13), and it was primarily due to the recombination of capsule and LPS biosynthesis loci, which have been defined as recombination hot spots subject to strong diversifying selection in *K. pneumoniae*[8,20]. From an epidemiological standpoint, it is highly interesting to pinpoint the selective pressures for the prevalence of O2v1:KL64. Consistent with our data that O2 was prevalent in ST11 and ST15, a previous study showed that the O2 serotype was dominant in CRKP (50%) in the last two decades, especially in another epidemic clone ST258[21]. The prevalence of O2 antigen supposedly correlates with a paucity of anti-O2 antibodies in human B cell repertoires[21]. Additional studies reported that patients with O2v1:KL64-CRKP BSI had a significantly higher 30-day mortality rate and a higher sepsis/septic shock incidence rate[4,22]. These suggest that the emergence of O2v1:KL64 is associated with host susceptibility resulting in enhanced pathogenicity by evasion of innate host defense. Hence, the identified subclonal shift within CRKP-ST11 is likely to cause

challenges to the current infection control measurements and treatment strategies.

Clonal replacement has been reported in multiple notorious MDR pathogens, and one of the more well-known examples is methicillin-resistant *Staphylococcus aureus* (MRSA)[23,24]. Evolved resistance and virulence-associated with a set of genetic alterations have been linked to the clonal replacement of MRSA[25–27]. In this study, we revealed that the epidemic success of O2v1:KL64 is associated with the expansion of a hypervirulent lineage with the capture of a virulence plasmid. The emergent O2v1:KL64-hvKP isolates have been sporadically reported in several surveillance surveys[28,29]; however, none of them has systematically investigated the epidemiological and evolutionary characterizations of the lineage. Numerous studies have demonstrated that virulence plasmids can promote the pathogenicity of *K. pneumoniae*[30–32]. Indeed, we here found that the O2v1:KL64-hvKP obtained enhanced phagocytosis resistance compared with those isolates without virulence plasmids. Our previous study showed that virulence plasmids promoted pathogen invasion and subsequent clinical infection, and O2v1:KL64-hvKP displayed enhanced resistance to neutrophil killing[4]. Together, their data confirm that O2v1:KL64-hvKP has evolved to be more virulent through capturing virulence plasmids. The emergence of O2v1:KL64-hvKP is additionally associated with the capture of MDR plasmids. A strong correlation was found between five ARGs [i.e., $bla_{LAP-2}$, *dfrA*-like, *qnr*-like, *sul*-like, and *tet(A)*] and O2v1:KL64-hvKP but not OL101:KL47 (Fig. 5c), which is due to the enrichment of an IncFII MDR plasmid. Compared with the other subpopulations, the O2v1:KL64-hvKP displayed the highest resistance to tetracycline and sulfamethoxazole-trimethoprim (Fig. 5b), suggesting that these two drugs might have been involved in the prevalence of O2v1:KL64-hvKP. O2v1:KL64 and OL101:KL47 showed comparable resistance to ciprofloxacin due to mutations in *gyrA* or *parC*, while additional loss and gain of *oqxAB* and *qnr*-like genes, respectively, was found in O2v1:KL64. Of note, *oqxAB* have widely been considered to be core genes of *K. pneumoniae*[6], but our results showed that they were lost in most of the O2v1:KL64 isolates, possibly by IS-mediated and/or recombination mechanisms (Supplementary Fig. 15). As a multidrug efflux pump, OqxAB confers low to intermediate resistance to several antibiotics (e.g., quinoxalines, quinolones, tigecycline, and nitrofurantoin), detergents and disinfectants. Loss of OqxAB may affect the drug resistance of O2v1:KL64. Whether such a loss could confer evolutionary advantages to the epidemic success needs to be studied further.

In bacteria, recombination, horizontal gene transfer, and mutations are recognized as major sources of the genetic variations introduced into a population. As found in other prevalent MDR clones (e.g., ST258 and ST15)[8], the genomes of ST11 have been shaped by frequent recombination events. In particular, our analysis suggested that the contribution of recombination to the genetic variations was higher in O2v1:KL64 than in OL101:KL47, and we further demonstrated that a point mutation occurring in *recC* of O2v1:KL64 conferred more recombination proficiency, which was unique to O2v1:KL64 isolated in China. It is known that *recC*, together with *recB* and *recD*, encodes an ATP-dependent nuclease, called RecBCD enzyme, and the RecBCD-dependent pathway is the primary mechanism of homologous recombination and repair of linear double-strand DNA in *E. coli*. The structure and function of the RecBCD enzyme is regulated by Chi sites (5′-GCTGGTGG-3′) to stimulate recombination[33]. Current evidence suggests that the RecC subunit recognizes Chi in the 3′ tunnel[34], and a 35-kDa C-terminal domain of RecC is required for interaction with the RecD protein, a prerequisite for responsiveness to Chi[35]. Point mutations in the C-terminal domain of RecC indirectly prevent RecD from associating with RecBC[35], resulting in the "double-dagger" phenotype: recombination proficiency that is independent of Chi and the absence of nuclease activities[36]. Of note, the point mutation (His935Arg)

occurring in RecC of O2v1:KL64 is located in the C-terminal domain; we, therefore, infer that it may affect the interaction with the RecD and responsiveness to Chi to stimulate recombination. Further, It was shown that mutations that inactivate the *recB* or *recC* gene lead to defects in multiple biological function, including conjugational, transductional, and phage recombination; a loss of SOS induction; sensitivity to DNA-damaging agents that cause DSBs; and low cell viability[37]. We, therefore, propose that the O2v1:KL64 population carrying the *recC* mutation had become more diverse through more frequent recombination and by capturing more MGEs as detected in this study, and O2v1:KL64-hvKP was selected for and become prevalent in a short time scale.

In addition, horizontal gene transfer was identified as another driving force for the diversification in ST11, since a significantly higher load of MGEs, including plasmids, ISs, and prophages, was detected in O2v1:KL64 compared to OL101:KL47 (Supplementary Fig. 5a–d). It is known that phage-induced selective pressures play a critical role in the population diversity of bacteria[13], and we found that the prophage load was significantly higher in O2v1:KL64 than in OL101:KL47 due to multiple gain and loss events (Fig. 4c). These findings suggest that the two subclones might have been exposed heterogeneously to different types of phages. In particular, our analysis pinpoints that *Siphoviridae* prophages contributed significantly to the high prophage load in O2v1:KL64, which probably resulted from a single acquisition event through the vertical model (Fig. 4d). Similarly, the large-scale expansion of plasmids and ISs within O2v1:KL64 was also linked to a single acquisition event followed by vertical inheritance (Fig. 4b, c). The vertical transmission of these MGEs suggests that they may have conferred evolutionary advantages to O2v1:KL64, which could be further supported by the identified "successful" genotypes carrying these MGEs (Fig. 6). Given that ST11 genomes harbor more MGEs than non-ST11 in our collection (Supplementary Fig. 6), we propose that accumulation of prophages and plasmids could be one of the vital factors for the epidemic success of successful clones. Despite the benefits that cargo genes (e.g., AMRs) carried by MGEs can provide, the introduction of novel MGEs in a pre-existing, well-tuned genetic background would incur a fitness cost, and the maintenance of MGEs in host cells requires a balance of the costs and benefits to the host[38], e.g., minimized over time by selection[39]. It would be interesting to explore the underlying mechanisms employed by O2v1:KL64 to fine-tune the fitness costs introduced by these MGEs. Additionally, future studies should involve examining the biological function of these genetic elements to understand their role in the population-level success of O2v1:KL64.

Of greater concern, using an SNP-based transmission tracking method, we revealed that the prevalence of O2v1:KL64-hvKP was driven by an altered transmission pattern. The accuracy of SNP-based transmission tracking methods has been recently evidenced by numerous studies using large datasets for various pathogens, like MRSA and CRKP[7,16,40]. In particular, we here used a range of SNP cutoffs to avoid any bias that could be caused by a single cutoff. Indeed, all SNP cutoffs used in our analysis generated a consistent transmission trend, demonstrating the reliability and validity of our results. The spread of O2v1:KL64-hvKP was mainly driven by an intra-hospital transmission before 2018, which could be attributed to evolutionary advantages conferred by a set of genetic alterations as identified here. With the increasing size of O2v1:KL64-hvKP over time, inter-hospital patient transfers may have further facilitated the spread of the subpopulation across the country, leading to a subsequent switch to inter-hospital transmission. More metadata is needed to uncover the reasons for changes in transmission mode in the future. Our findings highlight the necessity of tailoring the current infection control measures (e.g., active screening of inter-hospital transferred patients with a history of CRKP) to prevent the dissemination of O2v1:KL64-hvKP in China.

In summary, we have shown here that subclonal replacement within CRKP-ST11 has been driven by the expansion of O2v1:KL64 within a 6-year period in China. The epidemic success of O2v1:KL64 is associated with the emergence and dissemination of a subpopulation associated with a repertoire of genetic alterations, and of greater concern, with the enhanced inter-hospital transmission. Collectively, our study highlights that public health efforts should focus on genomic surveillance to identify high-risk clones and subclonal expansions early in the course of an epidemic to potentiate targeted control strategies.

## Methods

### Dataset

A total of 4635 non-duplicate isolates were collected in the framework of national surveillance for blood isolates (Blood Bacterial Resistant Investigation Collaborative System, BRICS) between January 2014 and December 2019 in China (Fig. 1a). Only the first blood isolate of each species per patient was eligible over the full study period. All participating hospitals sent their isolates to the central laboratory quarterly. Species identification was performed by matrix-assisted laser desorption/ionization time-of-flight mass spectrometry (MALDI-TOF/MS) (Bruker Daltonik GmbH, Bremen, Germany). The increase in the number of isolates over time was partly due to improvements in quality control by participating hospitals during the surveillance period. The sample collection protocol was approved by the institutional review board of the First Affiliated Hospital of Zhejiang University in China.

### Antimicrobial susceptibility testing

Antimicrobial susceptibility testing was initially performed using a VITEK-2 system (bioMérieux, Lyon, France) in the sentinel hospitals and was further confirmed by the agar dilution and/or broth microdilution method in our laboratory. Results were interpreted according to the Clinical and Laboratory Standards Institute[41] and European Committee on Antimicrobial Susceptibility Testing v.10.0 (http://www.eucast.org/clinical_breakpoints/). Carbapenem resistance was defined as a minimum inhibitory concentration (MIC) of ≥4 mg/L for imipenem or meropenem.

### Whole-genome sequencing and quality control analysis

Genomic DNA from 794 CRKP isolates was extracted using Gentra Puregene Yeast/Bact. Kit (Qiagen, San Francisco/Bay Area, CA, USA). The genomes were sequenced using an Illumina Novaseq 6000 system (Illumina, San Diego, United States) with $2 \times 150$-bp paired-end libraries. Raw reads were trimmed using Trimmomatic v0.33[42] and then assembled using SPAdes v3.12.0[43]. We performed long-read sequencing on representative isolates using a Nanopore PromethION platform (Nanopore, Oxford, UK) following a 10-Kbp library protocol. A hybrid assembly was generated by using Unicycler 0.4.0[44] with short and long reads. QUAST v4.6.0[45] was used to generate assembly statistics. Species were determined using FastANI v1.33 (https://github.com/ParBLiSS/FastANI), with a cut-off of 95%. The assemblies were annotated using Prokka v1.14.6[46].

### Identification of STs and K/O-type

The STs were assigned using Kleborate v2.0.1[47], and the K/O-type was determined using Kaptive v1.0[48] from the de novo assembly.

### Identification of ARGs, VFs, Inc-type, and MGEs

ARGs were detected using Abricate v1.0.1 (https://github.com/tseemann/abricate) with the ARG database ResFinder v4.0[49]. The VFs of *K. pneumoniae* were downloaded from the *K. pneumoniae* BIGSdb[50] and Virulence Factor Database 2019 (VFDB)[51], and the presence of VFs were detected using BLASTp v2.6.0 (identity ≥70%) with the custom VF database containing 154 VF genes

(Supplementary Dataset 3). Replicon typing was performed using Abricate v1.0.1 with the PlasmidFinder database[52]. The presence of integrons was detected using IntegronFinder v1.5.1[53]. IS copy numbers were estimated with the TPM calling function of TPMCalculator v0.0.3[54] and corrected by the TPM of *gapA* (a housekeeping gene). The prophages were detected and classified using phigaro v2.2.6[55]. The ancestral state of each MGEs was reconstructed with maximum likelihood using the fastAnc function in the R package phytools v.0.4-98[56].

### Recombination detection

Recombination analysis was performed using Gubbins v2.2[57]. The Gubbins output files were used to calculate r/m and mean recombination counts per base, calculated over non-overlapping 1000 bp windows. Pairwise nucleotide divergence between subclone-specific core-genome regions was calculated for each pair of genomes within a subclone before and after the removal of putative recombinant regions.

### Knockout, replacement, and complementation of *recC*

A representative O2v1:KL64 isolate KP37485 was used for the genetic manipulation. All primers used in this study were listed in Supplementary Dataset 10. For the knockout construction, the *recC*-spacer DNA fragment was cloned into a pSGKP-apr vector[58], and the synthetic oligonucleotide with 45 nt for each homology extension of the target gene was used as a donor template. Both the pSGKP-*recC*-spacer plasmid and the donor template DNA were transformed into pCasKP-harboring KP37485 by electroporation, and the integration was selected on LB plates containing 5% sucrose at 37 °C. For the gene replacement, a linear fragment containing a *cat* gene with its native promoter from plasmid pKD3[59] between the 45 nt each homology extension of *recC* was amplified and used as a donor template, and the resulting KP37485-*recC::cat*-pCasKP was selected on LB plates containing 5% sucrose plus 25 mg/L chloramphenicol and 30 mg/L apramycin at 30 °C. The synthetic oligonucleotide with $recC_{OL101:KL47}$ as well as its 45 nt each homology extension was used as the donor template and were transformed into KP37485-*recC::cat*-pCasKP with pSGKP-cat-spacer plasmid. The integration was selected as described in knockout construction, in addition, to being selected by chloramphenicol sensitivity. For the complementation, genes were amplified by PCR and cloned into vector pBAD33[60]. The resulting plasmids were introduced into kp37485Δ*recC* via electroporation. PCR and DNA sequencing were used to confirm the final constructions.

### Mitomycin C resistance assay

Recombinational repair of DNA damage mediated by the Rec-dependent pathway is known to be a primary strategy to protect bacteria from DNA-damaging agents (e.g., mitomycin C, ethidium bromide, and UV); therefore, these agents have been used extensively as indicators of recombination proficiency[11,35,61]. Mitomycin C resistance was assessed as previously described with slight modifications[61]. *K. pneumoniae* isolates were grown overnight in LB broth and inoculated into the fresh broth with indicated agents. Strains harboring pBAD33 and its derivative plasmids were grown in LB broth with 0.2% (w/v) L-arabinose and 25 mg/L chloramphenicol. The bacteria were harvested at the late logarithmic phase and suspended in phosphate-buffered saline (PBS), to $5 \times 10^8$ colony-forming units (cfu) per milliliter. Cell suspensions were treated with mitomycin C at indicated concentrations for 1 and 3 h, serially diluted in PBS, spread on LB plates, and incubated overnight at 37 °C. Bacterial treated with PBS was used as the negative control. The survival ratio was calculated as follows: (CFU of mitomycin C treatment culture/CFU of PBS-treated culture) × 100%.

## Recombination assays

The suicide vector pRE118, carrying the upstream and downstream fragment of $bla_{KPC-2}$ of the KP37485 and an apramycin resistance gene *apmR*, was generated and electroporated into tested hosts. Recombinants were selected on 50 mg/L apramycin-containing plates and further confirmed by PCR. Recombination frequency was calculated as the number of recombinants/the number of recipients.

## Phagocytosis assay

Phagocytosis assay was performed as previously described in ref. 62. In brief, THP-1 cells were differentiated at $1 \times 10^6$ cells/well in a 12-well plate. Bacteria in the mid-log phase were harvested by centrifugation (5 min, 6000 rpm, 24 °C), resuspended in 1×PBS, and adjusted to $5 \times 10^8$ CFU/mL. Infections were performed using a multiplicity of infection (MOI) of 50 bacteria per cell. To synchronize the infection, plates were centrifuged at 200×*g* for 5 min and incubated at 37 °C in a humidified 5% $CO_2$ atmosphere. After 1 h of contact, cells were washed twice with PBS and cultured with RPMI 1640 containing 10% FBS and gentamicin (100 mg/L). To determine the bacterial load in the cell, the cells were washed twice with PBS and lysed with 0.5% saponin in PBS for 10 min at room temperature. Serial dilutions were plated on LB to quantify the number of intracellular bacteria. All experiments were carried out with triplicate samples on at least three independent occasions.

## Plasmid analysis

Short-read assemblies were blasted against reference plasmids using BLASTn v2.9.0[63] to determine the length of the reference plasmid sequence present across isolates. Replicon typing and clustering were performed using MOB-suite[15]. Representative plasmids were circularized using PCR and Sanger sequencing. The Artemis Comparison Tool v13.0.0[64] and/or BRIG[65] was used to compare and visualize structural variation between two or more sequences. The heatmap showing the percentage of aligned regions between pairs of virulence plasmids was generated using the "pheatmap" package (v1.0.12) in R v4.1.1 (https://www.r-project.org/).

## Phylogenetic analysis of Chinese ST11 isolates collected in this study

Trimmed sequencing reads of 646 ST11 isolates collected in this study were mapped to a *K. pneumoniae* ST11 reference genome KP16932 (accession no. QVAN00000000)[4] using BWA mem 62 v0.7.10-r789 (default parameters). Mapped reads were then cleaned and sorted using the SAMtools suite v1.7[66]. Reads were realigned against the reference using GATK v3.7[67] by creating targets for realignment (RealignerTargetCreator) and performing realignment (IndelRealigner). Removal of optical duplicates was completed using Picard v2.10.1-SNAPSHOT (https://broadinstitute.github.io/picard/). Sequence variants were called using Bcftools v1.9-80 (http://samtools.github.io/bcftools) to generate a reference-based pseudogenome for each genome with greater than 10× depth. High-quality pseudogenomes were concatenated (plasmid sequences excluded) before Gubbins v2.2[57] was used to remove recombinant regions and invariable sites. Forty ST258 genomes (indicated by the triangle) retrieved from GenBank were included to establish the root of the ST11 tree. The resultant multiple sequence alignment of reference-based pseudogenomes (4460 variant sites) was used to infer a maximum-likelihood phylogeny using RAxML-ng v0.6.0[68] with 100 bootstrap replicates to assess support.

## Phylogenetic and BEAST analysis of a global collection of ST11 isolates

All available genomes of ST11 ($n = 329$) in GenBank retrieved as of March 1, 2021, were included in this analysis (Supplementary Dataset 9). Snippy v4.6.0 (https://github.com/tseemann/snippy) was used to align the 975 genomes to the reference genome KP16932 to generate the alignment of core-genome SNPs. SNPs located in recombination regions were detected by Gubbins v2.2[57]. The resultant recombination-free core-genome SNPs (3027 variant sites) were used to infer a maximum-likelihood phylogeny using RAxML-ng v0.6.0[68] with a GTR model and gamma correction, and 100 bootstrap replicates were performed to assess support. A chronogram was produced using Bayesian phylogenetic inference. To reduce computation time, 147 genomes, including all KLs, were chosen for the analysis. Analysis of temporal molecular evolutionary signals for the dataset was conducted using TempEst v1.5[69]. A recombination-free core-genome alignment (1995 SNPs) was created using Snippy v4.6.0. BEAST v1.10.4[70] was used to create and execute three independent chains of length 250,000,000 with 10% burn-in, logging every 25,000 and accounting for invariant sites. We included the prior assumptions of a coalescent Bayesian skyline model for population growth, and a relaxed log normal clock rate to account for rate heterogeneity amongst branches. Convergence of the Markov chain Monte Carlo (MCMC) chain was inspected in Tracer v1.7.2[71], with all parameter effective sampling sizes being >200. The maximum clade credibility (MCC) tree under each model was generated in TreeAnnotator and plotted in FigTree v1.4.4 (https://github.com/rambaut/figtree).

## Intra- and inter-facility transmission analysis

Recombination-filtered core-genome SNPs of each dataset were generated as described above to calculate the pairwise SNP distance matrices. For each pair of unique patient–facility combinations, only the most closely related isolate pair was included as the pairwise genetic distance. A range of thresholds was used here to identify recent intra- and inter-facility transmission pairs using pairwise distances. The minimum threshold was determined as 14 SNPs ($2 \times 5,716,474 \times 1.2256 \times 10^{-6}$) using reference genome length (5,716,474 base pairs) and mutation rate ($1.2256 \times 10^{-6}$ mutations per base pair per year estimated in this study), and the maximum threshold was defined as 25 SNPs according to recent studies of CRKP transmission[7,16–18]. This analysis was performed using the regentrans package[72] in R v4.1.1.

## Statistical analysis

The Wilcoxon rank-sum tests and Chi-square tests were performed for pairwise group comparisons of MGEs, VFs, ARGs, and Pearson's Coefficient. The student's *t*-tests were used for pairwise group comparisons in phenotypical assays. The Mann–Kendall test was used to test for transmission trends. *P* values <0.05 were considered significant. All statistical analyses were implemented in R v4.1.1.

## Reporting summary

Further information on research design is available in the Nature Portfolio Reporting Summary linked to this article.

# Data availability

All assembled Illumina sequence data have been deposited in GenBank under the BioProject accession number PRJNA778807. Individual accession numbers are also available in Supplementary Dataset 1. Source data are provided with this paper.

# Code availability

The custom code used in this study is freely available at https://github.com/xuechunxu/CRKP_ST11_KL64 (https://doi.org/10.5281/zenodo.7804081).

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

## Acknowledgements

We thank Jingjie Song and Yang Liu for their technical support of bioinformatics. We also thank Jinru Ji and Chaoqun Ying for preparing the metadata of the collection. This work was funded by the National Key R&D Program of China under award numbers 2021YFC2300300 (to K.Z. and Y.X.) and 2022YFE0103200 (to K.Z.), the National Natural Science Foundation of China under award numbers 81902030 (to T.X.), 81971984 (to Y.X.), and 82172330 (to K.Z.), China Postdoctoral Science Foundation under award number 2022M712187 (to C.-X.X.), Shenzhen Basic Research Key projects under award number JCYJ20200109144220704 (to K.Z.), Shenzhen Basic Research projects under award number JCYJ20190807144409307 (to K.Z.), the Fundamental Research Funds for the Central Universities under award number 2022ZFJH003 (to Y.X.), and Research Project of Jinan Microecological Biomedicine Shandong Laboratory under award number JNL-2022027C (to Y.X.).

## Author contributions

K.Z., Y.X., and K.E.H. conceived the project and revised the manuscript. The BRICS Working Group collected the bacterial isolates and epidemiological data and performed preliminary laboratory analyses. K.Z., C.-X.X., P.S., Y.C., K.L.W., and M.M.C.L. performed the data analysis. T.X., S.W., J.L., H.L., and P.S. performed the experiments. K.Z. wrote the manuscript. K.Z. and Y.X. is the guarantor of this work and, as such, had full access to all of the data in the study and takes responsibility for the integrity of the data and the accuracy of the data analysis. All authors read and approved the final version of the manuscript.

## Competing interests

The authors declare no competing interests.

## Additional information

[1]Shenzhen Institute of Respiratory Diseases, Shenzhen People's Hospital (Second Clinical Medical College, Jinan University; The First Affiliated Hospital, Southern University of Science and Technology), Shenzhen 518020, China. [2]State Key Laboratory for Diagnosis and Treatment of Infectious Diseases, National Clinical Research Center for Infectious Diseases, National Medical Center for Infectious Diseases, Collaborative Innovation Center for Diagnosis and Treatment of Infectious Diseases, The First Affiliated Hospital, Zhejiang University School of Medicine, Hangzhou 310003, China. [3]Department of Infectious Diseases, Central Clinical School, Monash University, Melbourne, VIC 3004, Australia. [4]Department of Clinical Laboratory, Shenzhen People's Hospital, Shenzhen, China. [5]Department of Infection Biology, Faculty of Infectious and Tropical Diseases, London School of Hygiene and Tropical Medicine, London WC1E 7HT, UK. [60]These authors contributed equally: Kai Zhou, Chun-Xu Xue, Tingting Xu, Ping Shen. ✉e-mail: zhouk@mail.sustech.edu.cn; xiaoyonghong@zju.edu.cn

## the BRICS Working Group

Yunbo Chen[6], Hui Ding[7], Yongyun Liu[8], Haifeng Mao[9], Ying Huang[10], Zhenghai Yang[11], Yuanyuan Dai[12], Guolin Liao[13], Lisha Zhu[14], Liping Zhang[15], Yanhong Li[16], Hongyun Xu[17], Junmin Cao[18], Baohua Zhang[19], Liang Guo[20], Haixin Dong[21], Shuyan Hu[22], Sijin Man[23], Lu Wang[24], Zhixiang Liao[25], Rong Xu[26], Dan Liu[27], Yan Jin[28], Yizheng Zhou[29], Yiqun Liao[30], Fenghong Chen[31], Beiqing Gu[32], Jiliang Wang[33], Jinhua Liang[34], Lin Zheng[35], Aiyun Li[36], Jilu Shen[37], Yinqiao Dong[38], Lixia Zhang[39], Hongxia Hu[40], Bo Quan[41], Wencheng Zhu[42], Kunpeng Liang[43], Qiang Liu[44], Shifu Wang[45], Xiaoping Yan[46], Jiangbang Kang[47], Xiusan Xia[48], Lan Ma[49], Li Sun[50], Liang Luan[51], Jianzhong Wang[52], Haoyun Lin[53], Zhuo Li[54], Dengyan Qiao[55], Lin Zhang[56], Chuandan Wan[57], Xiaoyan Qi[58] & Fei Du[59]

[6]The First Affiliated Hospital of Zhejiang University, Hangzhou, China. [7]Lishui City Central Hospital, Lishui, China. [8]Affiliated Hospital of Binzhou Medical College, Binzhou, China. [9]the First People's Hospital of Lianyungang, Lianyungang, China. [10]First Affiliated Hospital of Anhui Medical University, Hefei, China. [11]Yijishan Hospital of Wannan Medical College, Wuhu, China. [12]Anhui Provincial Hospital, Hefei, China. [13]Wuhan Puren Hospital, Wuhan, China. [14]The First People's Hospital of Jingzhou, Jingzhou, China. [15]People's Hospital of Ningxia Hui Autonomous Region, Yinchuan, China. [16]Anyang District Hospital of Henan Province, Anyang, China. [17]The Second People's Hospital of Yunnan Province, Kunming, China. [18]Zhejiang Provincial Hospital of Traditional Chinese Medicine, Hangzhou, China. [19]People's Hospital of Huangshan City, Huangshan, China. [20]Mindong Hospital of Ningde City, Ningde, China. [21]The Affiliated Hospital of Jining Medical University, Jining, China. [22]People's Hospital of Qingyang, Qingyang, China. [23]Tengzhou Centre People's Hospital, Tengzhou, China. [24]Lu'an People's Hospital, Lu'an, China. [25]Xinjiang Uygur Autonomous Region Youyi Hospital, Urumqi, China. [26]People's Hospital of Yichun City, Yichun, China. [27]Jiujiang First People's Hospital, Jiujiang, China. [28]Shandong Provincial Hospital, Jinan, China. [29]Jingzhou Central Hospital, Jingzhou, China. [30]the First Affiliated Hospital of Gannan Medical University, Ganzhou, China. [31]The First Hospital of Putian City, Putian, China. [32]People's Hospital of Haining City, Haining, China. [33]Shengli Oilfield Central Hospital, Dongying, China. [34]The Affiliated Hongqi Hospital of Mudanjiang Medicine College, Mudanjiang, China. [35]The Affiliated Hospital of Ningbo Medical School, Ningbo, China. [36]Women's Hospital, Zhejiang University School of Medicine, Hangzhou, China. [37]The Fourth Affiliated Hospital of Anhui Medical University, Hefei, China. [38]Tianchang City People's Hospital, Chuzhou, China. [39]Shanxi Provincial People's Hospital, Taiyuan, China. [40]The First Affiliated Hospital of Henan University of Science and Technology, Luoyang, China. [41]The Second People's Hospital of Jingzhou, Jingzhou, China. [42]Lu'an Civily Hospital, Lu'an, China. [43]The Second Affiliated Hospital of Bengbu Medicine College, Bengbu, China. [44]Huaihe Hospital of Henan University, Kaifeng, China. [45]Qilu Children's Hospital of Shandong University, Jinan, China. [46]Zigong Third People's Hospital, Zigong, China. [47]the Second Hospital of Shanxi Medical University, Taiyuan, China. [48]the People's Hospital of Lujiang, Hefei, China. [49]the First People's Hospital of Jiayuguan, Jiayuguan, China. [50]The Third Hospital of Hefei, Hefei, China. [51]General Hospital of Northern Theater Command, Shenyang, China. [52]Xingang Hospital of Xinyu, Xinyu, China. [53]Shenzhen People's Hospital, Shenzhen, China. [54]The First Affiliated Hospital of Xi'an Medical University, Xi'an, China. [55]Gansu Provincial Hospital of Traditional Chinese Medicine, Lanzhou, China. [56]First People's Hospital of Chenzhou, Chenzhou, China. [57]Changshu Medicine Examination Institute, Changshu, China. [58]Women and Children's Hospital of Jin'an District, Lu'an, China. [59]Hubin Hospital of Hefei, Hefei, China.

