## [Peer review file · Nature Communications]

REVIEWER COMMENTS

Reviewer #1 (Remarks to the Author):

The manuscript “A point mutation in recC promotes subclonal replacement of carbapenem-resistant *Klebsiella pneumoniae* ST11 in China” describes the genomic diversity of CRKP blood isolates in China from 2014 to 2019, and demonstrates that the ST11 subclone OL101:KL47 was replaced over time by subclone O2v1:KL64. The manuscript details multiple factors that likely contributed to the success of O2v1:KL64, including a point mutation in recC that increased recombination, and a hypervirulent sublineage with increased resistance to phagocytosis and certain antibiotics and heavy metals. Overall, the manuscript is well-written and provides a thorough description of ST11 CRKP blood isolates in China during the study period and important insight into the factors that contribute to the success of a particular subclone of ST11 CRKP.

Title: Correct the spelling of “pneumoniae” to “pneumoniae”.

Lines 40-41: What is meant by enhanced inter-hospital transmission? Is this saying that the sublineage was associated with more inter-hospital transmission?

Lines 88-89: How much variation is there among the blaKPCs identified? Were particular blaKPCs more prevalent?

Line 319: What is meant by “fluctuation”?

Lines 398-399: Please clarify what is meant by environmental selection.

Line 441: Change “measurements” to “measures”.

Lines 441-442: What are potential changes to infection control measure that would prevent the spread of the O2v1:KL64-hvKP sub-clone?

Supplementary Table 9: It would be helpful to have these represented as percentages (as in Supplementary Table 10).

Lines 618-620: Have the sequence read files for each isolate been deposited?

Supplemental Figure S8: Is the row/column order the same top to bottom and left to right?

Reviewer #2 (Remarks to the Author):

This study comprehensively characterized the recent genomic epidemiology of ST11 CRKP in China. It is an extension of a prior report of the ST11 KL64 subclone replacing KL47 by the same group. This new analysis encompasses 794 bloodstream isolates from multiple centers compared to 154 isolates from a single center in the prior study. It confirms the prior observation of the emergence of this subclone and subsequent expansion, this time in a larger geographic region (although the prior study had also reported the wider geographical distribution of this subclone). The key observation is that recombination was a major driver for the evolution of O2v1:KL64, and found an association of higher recombination frequency with a mutation in *recC*. This was further supported by genetic validation studies deleting the *recC* gene. Additional data highlight the complex mobile genetic repertoire of the subclone that might have provided additional advantages though these associations are less clear.

Major comments:

1. Figure 1 provides a good summary of the available samples. It appears though that the overall number of Kp blood stream samples (and not just CRKP) went up significantly during the study period. This may be due to more efficient sample collection; please discuss selection criteria in more detail (lines 78 – 80).
2. Only bloodstream isolates were included here. Do the authors have data on non-BSI isolates from the same time period, and are similar trends of subclonal replacement present in CRKP isolates from other sites?
3. All genetic validation experiments for *recC* were carried out in KP37485. How was this isolate selected for these studies?
4. Throughout the results it would be helpful to clearly state which observations were made previously and were reported in the EID paper by the same group and how the current study expands on these. For example, the *rmpA* and pVir-KP16932 plasmid were also reported previously.
5. Lines 235 – 239: Please provide more details in this paragraph on number of isolates and repeats. Also, the subheading of this paragraph is somewhat misleading as the data do not support an overall increased phagocytosis for the subclone compared to others.

6. The sections metal and antibiotics resistance are highly speculative. Why the authors are careful in their language there is no functional validation or linkage with drug use or other data that support some of these statements. This section could be significantly reduced.

7. O2v1:KL64 has a larger mobile genetic repertoire. While this may confer some of the functional advantages implied in the text what is the evidence for these selective pressures? In addition, what is the potential cost associated with maintaining this large accessory genome?

8. The study reports increased inter-facility transmission of CRKP. However, this may in part be due to the higher number of samples available for analyses. Please include this in the discussion (line 429 onwards).

9. Line 356 – it would be better to indicate that the current study expanded on the prior findings regarding this clone.

10. Line 419 – 428 on the mobile repertoire is highly speculative and not further validated here; as above please consider pros and cons of the high load.

Minor comments:

- Lines 35 – 40 – please simplify this very long sentence.

Reviewer #3 (Remarks to the Author):

This manuscript reports on a large survey of CR-KP in China and its extensive genomic and phenotypic analysis. The manuscript is extremely rich as it includes multiple aspects from epidemiological trends to comparisons of isolates from other geographic origin (global), analyses of recombination and pangenomic/MGE features, phenotypic data of selected isolates, to the experimental study of a particular mutation in *recC*. This reviewer was a bit overwhelmed by the density of the data even though I found the manuscript generally well written and concise, and most of the main points are very relevant to understanding this emergence, and interesting from a biological viewpoint.

As is the case when a manuscript is so rich, it is hard for the authors to go into every detail of each of the aspects of the work, and for the reviewer to dive deeply into each section and be proficient in each aspect. I have highlighted a few points of attention that came to mind at first-pass reading. My two main points concern first, the repeated inferences on the possible selective value of observed genomic variation, which to my mind should be discussed altogether, and very carefully given the physical linkage

of so many variable markers; and second, the study of recC mutation, where I am no expert but feel the mechanistic aspects of the observation are not at all discussed enough to convince the reader of a functional impact on the observed emergence (where is the mutation located on the recC protein, is it likely to affect an active/functional site, what type of recombination is RecC involved in, etc.)

Third, the novelty of the K64 clade emergence and its characteristics should also be discussed and put into the context of existing knowledge, as previous literature exists on this observation. For example, Yang et al (Emerg Microbes Infect. 2020) state: the emergence of one ST11-K64 Hv-CRKp strain was related to the acquisition of rmpA/rmpA2 genes and siderophore gene clusters; but this paper is not cited.

Specific points:

Figure 2: rooting of the tree is a very important aspect of the work to establish the evolutionary scenario. The authors should provide evidence that rooting with closely-related outgroups would provide a concordant picture of the ST11 evolution, especially with respect to defining the ancestral/early-diverging branches.

Figure 2 legend: subclones are not monophyletic, which is at odds with the classical definition of a clone, a group of individuals descending from a unique common ancestor. Generally speaking I would have issues with the definitions of subclones by their surface antigens, rather than based on phylogenetic relationships, but this is a minor point that can easily be amended.

oqxAB is considered a core gene in Kp. Please check or discuss the finding that it is apparently not conserved in ST11

Line 133: how is SNP divergence calculated (what is the denominator)

Line 149: please explain the link between recombination proficiency and mitomycin resistance

Line 151: "demonstrating that recC plays a similar role in K. pneumoniae to that in E. coli" seems too general sentence; you only demonstrated one effect of recC in Kp

Line 153: was the isogenic mutant checked for other mutations in the genome? As far as I understand, the mutant was not engineered to only modify the single nucleotide position in *recC*; please precise if there were other SNP differences in the introduced versus recipient *recC*, and possibly any other artefactually introduced difference observed after engineering the mutant.

Line 163: typo: 1,4407 K. pneumoniae genomes. Currently, there are more than 50 000 genomes of Kp in public repositories.

Line 164: the allele was found in 763 of 823 ST11 O2v1:KL64 isolates : that is interesting , as it suggests that *recC* mutation was acquired after the capsule switch event to K64. Could this event also play a role in recombination proficiency, on top of the *recC* mutation itself? Not sure you have the right controls here. Please indicate the position of the mutation (and of the reference strain 37485) in your tree in Figure 2.

Please also state clearly if the mutation was never observed outside of the K64 ST11 genomes.

Line 175: the statistical tests suppose an independence of observations; here the higher load of MGE is likely to derive from ancestral relationships and vertical descent (after a single acquisition or loss) within the K64 branch, so I would not consider these statistics as being meaningful. The difference in MGEs could simply result from a few acquisition events at the root of the ST11-K64 branch of just later. It would be more meaningful to map the acquisition or loss of the elements along the phylogeny and discuss the relative number of such single events.

Lines 186-188: same comment as just above; please identify unique acquisition events along the phylogeny. It would also provide a view whether the MGE acquisition is still ongoing, or rather ancestral in the K64 branch.

Line 192: “potential pathogenesis”, as this is genomic observation, rather than phenotypic

Line 209: suggesting that the expansion of O2v1:KL64 was driven by the prevalence of the hvKP population: as many inferences on selective value made elsewhere, it is very debatable whether such logical inference is correct; the HV genes could be hitchhiking with other selective markers rather than being the driver; TMP/STX resistance for example is also higher in this branch (Figure 5), a likely driver too.

Lines 200 and following, and Figure 2: is it possible to separate *rmpA2* and *rmpADC* in the analysis and results reporting? It is unclear from Fig 2 if it is *rmpA2*, or rather *rmpADC*, that is acquired/lost multiple times in the K64 branch.

Lines 210 and next: how do your plasmid types related to KpVP-1 and KpVP-2 described by Lam et al.?

Figure 5A: please specify which three isolates were used in each of the four columns, and how the 12 triplicates variation was, as this would reinforce your finding.

Line 240: typo for pVLPK (should be pLVPK)

Line 247: implying that *rmpA/A2*-positive O2v1:KL64 might have been selected by certain heavy metals: again, this hypothesis is very risky due to physical genetic association of these genes with others

Line 249: Isolates with sil showed a higher resistance to Ag⁺ (MIC = 0.06 mM) than those without sil (MIC = 0.04 mM): this looks like a very small difference; how to assess its significance? Lines 240-255 could probably be taken out of the work without much harm.

Line 268: again you make an explanatory hypothesis on the emergence, which is made prudently indeed, but might be simply wrong. What is the use of these antimicrobials in China/the studied hospitals? I would suggest to remove these hypotheses from the Results section and to discuss the relative selective advantages provided by all the unique genomic features of K64 branch jointly in the Discussion

regarding *oxqAB*, do you have long-read sequencing data supporting their absence/presence and what genetic mechanism (insertion, deletion, or rather, assembly artefact) is involved?

Line 288: where it likely conferred selective advantages: same remark as above

Regarding the structure of the Results section, I wonder if the different observations (HV, ARGs, MGEs etc) should not be placed before the *recC* observation/experimental work, which might provide some explanation to the previous observations

Line 291-292: Given that the IncFIB-type virulence plasmids and IncFII-type MDR plasmids confer evolutionary fitness to O2v1:KL64: Here you take for granted, without justification, what you previously put forward as simple hypotheses.

Lines 290-301: given the uncertainty in detecting individual genes in short-read assemblies I would be wary of defining so many genotypes precisely. This entire paragraph could be replaced by a discussion section on the existence of 'higher-risk' subtypes carrying a more complete set of genes of concern. Determining what is selective among these is an entirely different matter.

Lines 303-323: how was the transmission defined, is it based on existence of individual isolates from same/different hospitals within individual genetic clusters?

This entire paragraph should be placed much closer to the beginning of the Results section, as it would enable to define transmission groups/clades that you could use later when describing the gene content of specific tree branches.

Lines 324-347 should equally be placed much earlier, at the beginning of the Results section, as it would lower the repetitions with the description of the Chinese-only dataset and put it into its global context; the temporal evolution and timing of ancestral nodes is very interesting to have in mind when reading the rest of the Results.

The fact that another sub-branch independently acquired O2/K64 in Brazil (but elsewhere too it seems from figure S11) also suggests an advantage of this O/K combination. This seems the hypothesis that you put forward at the beginning of the Discussion; but then it is unclear how the other observations you made contribute to the emergence/selective advantage.

Line 399: pointing to the role of environmental selection in the prevalence of O2v1:KL64-hvKP. Here again, you suppose that all genes provide a selective advantage, although a single driver might be sufficient for each emerging sublineage. In this precise case, environmental selective pressure would suppose transmission between humans and the environment, but you reasoned about mostly hospital transmission in the previous sections.

Lines 429-442: the discussion on transmission seems to be too disconnected from the rest. If a selective advantage was acquired through MGEs, it means the genotype will spread better within hospitals, and then logically more between hospitals too. I don't think you can/need to distinguish advantages that enhance between- rather than within- hospital transmission. What about possible changing patterns of inter-hospital patient transfers that may have played a role?

Line 445: The epidemic success of O2v1:KL64 is promoted by the emergence and dissemination of a hypervirulence subpopulation : again, this is a hypothesis unduly presented as evidence/conclusion. There is a need to revise entirely the manuscript regarding the selective inferences of genomic (and to a lesser extent, even phenotypic, as they are from in-vitro experiments) observations.

In the Discussion section, it would be critical to discuss the mechanistic role of RecC, and on which type of recombination it acts. I am no expert here, but is recC really acting both on homologous recombination and on MGE mobility, including plasmids and phages, which use their own specific mobility machinery? Unfortunately, the recC finding and its implications in explaining the genomic patterns are not discussed enough, especially considering the title of the manuscript.

Figure S2: the recombined regions are relative to a reference. Possibly it would be clearer/more in line with the logic of the study, to highlight recombination in the O2/K64 branch, by using a K47 reference.

Figure S11: this pattern would suggest that a unique Chinese ST11 branch emerged from global diversity. Is this really the case? I had the inverse notion of emergence in China, followed by global spread. The pattern also suggests multiple independent origins of O2v1/KL64. How was tree constructed, what evolutionary model used, and was recombination purged? Possibly using only representative isolates of the diversity of the Chinese clade (as was done in Figure S12?) would be more appropriate, as the sheer number of genomes in this clade might have brought some phylogenetic artefact.

Figure S13: the root-to-tip /time correlation is really not convincing, visually. You should consider providing a demonstration of the temporal signal using a randomization test (i.e., showing that the

REVIEWER COMMENTS

Reviewer #1 (Remarks to the Author):

The manuscript "A point mutation in *recC* promotes subclonal replacement of carbapenem-resistant *Klebsiella pneumoniae* ST11 in China" describes the genomic diversity of CRKP blood isolates in China from 2014 to 2019, and demonstrates that the ST11 subclone OL101:KL47 was replaced over time by subclone O2v1:KL64. The manuscript details multiple factors that likely contributed to the success of O2v1:KL64, including a point mutation in *recC* that increased recombination, and a hypervirulent sublineage with increased resistance to phagocytosis and certain antibiotics and heavy metals. Overall, the manuscript is well-written and provides a thorough description of ST11 CRKP blood isolates in China during the study period and important insight into the factors that contribute to the success of a particular subclone of ST11 CRKP.

Reply: We appreciate the reviewer's time and valuable suggestions to help us improve our manuscript. We have taken all suggestions into consideration and revised the text accordingly.

Title: Correct the spelling of "pnuemoniae" to "pneumoniae".

Reply: We apologize for the typo, and it has been corrected in the text.

Lines 40-41: What is meant by enhanced inter-hospital transmission? Is this saying that the sublineage was associated with more inter-hospital transmission?

Reply: Yes, it is meant that the sublineage was associated with more inter-hospital transmission. To make it clearer, we have replaced "enhanced" by "more frequent".

Lines 88-89: How much variation is there among the *bla*KPCs identified? Were particular *bla*KPCs more prevalent?

Reply: Thank you for the comments. Two variations were found, including 709 *bla*_{KPC-2} and 3 *bla*_{KPC-3}. The information has been updated in the text.

Line 319: What is meant by "fluctuation"?

Reply: Indeed, we realized that "fluctuation" could lead to misunderstandings, therefore, the word has been replaced by "Relative stability".

Lines 398-399: Please clarify what is meant by environmental selection.

Reply: Thank you for the comments. Together with the comments of the other reviewers, the inference has been removed, and the discussion has been updated.

Line 441: Change "measurements" to "measures".

Reply: Thank you for the suggestion, and "measurements" has been replaced by "measures".

Lines 441-442: What are potential changes to infection control measure that would prevent

the spread of the O2v1:KL64-hvKP sub-clone?

Reply: Thank you for the comments. Our results showed that the spread of O2v1:KL64-hvKP was facilitated by more frequent inter-hospital transmissions, therefore, we suggest that some measures, such as active screening of inter-hospital transferred patients with a history of CRKP, should be routinely implemented to prevent the spread of the O2v1:KL64-hvKP sub-clone especially in China (active screening for CRKP is not a routine infection control measure in China). This information has been added to the text.

Supplementary Table 9: It would be helpful to have these represented as percentages (as in Supplementary Table 10).

Reply: Thank you for the comments. According to the comments of the other reviewers, the results of heavy metal resistance have been completely removed from the manuscript.

Lines 618-620: Have the sequence read files for each isolate been deposited?

Reply: We have deposited assembled genomes but not the sequence read files in NCBI. In fact, we had tried to upload the sequence read files several times, but the size of the sequence read files is too large, resulting in the failure of the upload along with unstable internet.

Supplemental Figure S8: Is the row/column order the same top to bottom and left to right?

Reply: Yes, the row/column was ordered top down and left to right. You may find the information in the figure legend (in the last bracket).

Reviewer #2 (Remarks to the Author):

This study comprehensively characterized the recent genomic epidemiology of ST11 CRKP in China. It is an extension of a prior report of the ST11 KL64 subclone replacing KL47 by the same group. This new analysis encompasses 794 bloodstream isolates from multiple centers compared to 154 isolates from a single center in the prior study. It confirms the prior observation of the emergence of this subclone and subsequent expansion, this time in a larger geographic region (although the prior study had also reported the wider geographical distribution of this subclone). The key observation is that recombination was a major driver for the evolution of O2v1:KL64, and found an association of higher recombination frequency with a mutation in recC. This was further supported by genetic validation studies deleting the recC gene. Additional data highlight the complex mobile genetic repertoire of the subclone that might have provided additional advantages though these associations are less clear.

Reply: We appreciate the reviewer's time and valuable suggestions to help us improve our manuscript. We have taken all suggestions into consideration and revised the text accordingly.

Major comments:

1. Figure 1 provides a good summary of the available samples. It appears though that the overall number of Kp blood stream samples (and not just CRKP) went up significantly during the study period. This may be due to more efficient sample collection; please discuss

selection criteria in more detail (lines 78 – 80).

Reply: Thank you for the comments. In the surveillance, only the first isolate of each species per patient was eligible over the full study period. All participating hospitals sent their isolates to the central laboratory quarterly. Indeed, sample collection efficiency was relatively lower at the beginning of the surveillance for some reasons, e.g. staff unfamiliarity with the procedures in the sentinel hospitals, but it became more and more efficient over time. The information have been updated in Methods.

2. Only bloodstream isolates were included here. Do the authors have data on non-BSI isolates from the same time period, and are similar trends of subclonal replacement present in CRKP isolates from other sites?

Reply: This is indeed an interesting question, however, the isolates included in this study were collected in the framework of national surveillance targeted at blood isolates. Therefore, non-BSI isolates were not available in the surveillance.

3. All genetic validation experiments for recC were carried out in KP37485. How was this isolate selected for these studies?

Reply: We used a CRISPR-Cas9-based system for the genetic manipulation, and selection markers for kanamycin and apramycin are available in this system. Therefore, we screened isolates susceptible to the two drugs in our collection, and KP37485 was the only one out of 70 isolates to meet the selection due to the extensively drug resistance of most CRKP isolates.

4. Throughout the results it would be helpful to clearly state which observations were made previously and were reported in the EID paper by the same group and how the current study expands on these. For example, the rmpA and pVir-KP16932 plasmid were also reported previously.

Reply: Thank you for the comments. We have updated the Results (section 'O2v1:KL64 is derived from OL101:KL47', and 'Emergence of a hypervirulent population by exclusively obtaining rmpA-positive virulence plasmids drives the expansion of O2v1:KL64'), and Discussion as suggested.

5. Lines 235 – 239: Please provide more details in this paragraph on number of isolates and repeats. Also, the subheading of this paragraph is somewhat misleading as the data do not support an overall increased phagocytosis for the subclone compared to others.

Reply: Thank you for the comments. The number of isolates has been added in the text, and the repeats have been shown in the legend of Figure 5A. The subheading of this paragraph has been revised as "Virulence plasmids are associated with significantly enhanced resistance to phagocytosis in O2v1:KL64".

6. The sections metal and antibiotics resistance are highly speculative. Why the authors are careful in their language there is no functional validation or linkage with drug use or other data that support some of these statements. This section could be significantly reduced.

Reply: Thank you for the comments. According to the reviewer and reviewer 3's suggestions,

the results of metal resistance have been completely removed.

7. O2v1:KL64 has a larger mobile genetic repertoire. While this may confer some of the functional advantages implied in the text what is the evidence for these selective pressures? In addition, what is the potential cost associated with maintaining this large accessory genome?

Reply: We fully agree with the reviewer's comments. Indeed, we have not direct evidence to support these selective pressures. Therefore, such descriptions have been removed from the manuscript throughout along with the other reviewers' suggestions.

The point with regard to the potential cost associated with maintaining this large accessory genome has been discussed as "Despite the benefits that cargo genes can provide, the introduction of novel MGEs in a pre-existing, well-tuned genetic background would produce a fitness cost, and the maintenance of MGEs in host cells requires a balance of the costs and benefits to the host. It would be interesting to explore in O2v1:KL64 whether fitness costs could be introduced by these MGEs, and if so, how they could be fine-tuned."

8. The study reports increased inter-facility transmission of CRKP. However, this may in part be due to the higher number of samples available for analyses. Please include this in the discussion (line 429 onwards).

Reply: Thank you for the comments. The discussion has been updated together with reviewer 3's suggestions.

9. Line 356 – it would be better to indicate that the current study expanded on the prior findings regarding this clone.

Reply: Thank you for the comments. The sentence has been updated as: "We previously detected a subclonal shift in CRKP-ST11 causing bloodstream infections in a tertiary hospital. In this study, we further demonstrated the subclonal shift over a 6-year national prevalence survey."

10. Line 419 – 428 on the mobile repertoire is highly speculative and not further validated here; as above please consider pros and cons of the high load.

Reply: Thank you for the comments. The point with regard to the potential cost associated with maintaining this large accessory genome has been discussed as "Despite the benefits that cargo genes can provide, the introduction of novel MGEs in a pre-existing, well-tuned genetic background would produce a fitness cost, and the maintenance of MGEs in host cells requires a balance of the costs and benefits to the host. It would be interesting to explore in O2v1:KL64 whether fitness costs could be introduced by these MGEs, and if so, how they could be fine-tuned."

Minor comments:

- Lines 35 – 40 – please simplify this very long sentence.

Reply: Thank you for the comments. The sentence has been rephrased.

Reviewer #3 (Remarks to the Author):

This manuscript reports on a large survey of CR-KP in China and its extensive genomic and phenotypic analysis. The manuscript is extremely rich as it includes multiple aspects from epidemiological trends to comparisons of isolates from other geographic origin (global), analyses of recombination and pangenomic/MGE features, phenotypic data of selected isolates, to the experimental study of a particular mutation in recC. This reviewer was a bit overwhelmed by the density of the data even though I found the manuscript generally well written and concise, and most of the main points are very relevant to understanding this emergence, and interesting from a biological viewpoint.

As is the case when a manuscript is so rich, it is hard for the authors to go into every detail of each of the aspects of the work, and for the reviewer to dive deeply into each section and be proficient in each aspect. I have highlighted a few points of attention that came to mind at first-pass reading. My two main points concern first, the repeated inferences on the possible selective value of observed genomic variation, which to my mind should be discussed altogether, and very carefully given the physical linkage of so many variable markers; and second, the study of recC mutation, where I am no expert but feel the mechanistic aspects of the observation are not at all discussed enough to convince the reader of a functional impact on the observed emergence (where is the mutation located on the recC protein, is it likely to affect an active/functional site, what type of recombination is RecC involved in, etc.)

Third, the novelty of the K64 clade emergence and its characteristics should also be discussed and put into the context of existing knowledge, as previous literature exists on this observation. For example, Yang et al (Emerg Microbes Infect. 2020) state: the emergence of one ST11-K64 Hv-CRKp strain was related to the acquisition of rmpA/rmpA2 genes and siderophore gene clusters; but this paper is not cited.

Reply: We appreciate the reviewer's time and valuable suggestions to help us improve our manuscript. We have taken all suggestions into consideration and revised the text accordingly.

Specific points:

Figure 2: rooting of the tree is a very important aspect of the work to establish the evolutionary scenario. The authors should provide evidence that rooting with closely-related outgroups would provide a concordant picture of the ST11 evolution, especially with respect to defining the ancestral/early-diverging branches.

Reply: Thank you for the comments. A few ST258 isolates were included as the outgroup to recalculate the phylogenetic tree, and the topology is consistent with the previous one. Figure 2 has been updated.

Figure 2 legend: subclones are not monophyletic, which is at odds with the classical definition

of a clone, a group of individuals descending from a unique common ancestor. Generally speaking I would have issues with the definitions of subclones by their surface antigens, rather than based on phylogenetic relationships, but this is a minor point that can easily be amended.

Reply: Thank you for the comments. "Subclone" has been replaced by "O/K combination".

oqxAB is considered a core gene in Kp. Please check or discuss the finding that it is apparently not conserved in ST11

Reply: We reanalyzed the data as suggested, and confirmed that oqxAB was lost in most of the KL64 isolates. This point has been discussed, and supplementary figure 15 has been added.

Line 133: how is SNP divergence calculated (what is the denominator)

Reply: As we mentioned in Methods (Pairwise nucleotide divergence between subclone-specific core genome regions was calculated for each pair of genomes within a subclone before and after removal of putative recombinant regions.), the denominator is the length of each subclone-specific core genome.

Line 149: please explain the link between recombination proficiency and mitomycin resistance

Reply: Thank you for the comments. "Recombinational repair of DNA damage mediated by Rec-dependent pathway is known to be a primary strategy to protect bacteria from DNA damaging agents (e.g. mitomycin C, ethidium bromide and UV), therefore, these agents have been used extensively as indicators of recombination proficiency." The information has been updated in Methods (section 'Mitomycin C resistance assay').

Line 151: "demonstrating that recC plays a similar role in K. pneumoniae to that in E. coli" seems too general sentence; you only demonstrated one effect of recC in Kp

Reply: Thank you for the comments. We have revised the sentence as "demonstrating that recC is involved in recombination in K. pneumoniae as that in E. coli".

Line 153: was the isogenic mutant checked for other mutations in the genome? As far as I understand, the mutant was not engineered to only modify the single nucleotide position in recC; please precise if there were other SNP differences in the introduced versus recipient recC, and possibly any other artefactually introduced difference observed after engineering the mutant.

Reply: We fully understand the reviewer's concern. The mutant was constructed through replacing the *recC* of O2v1:KL64 by that of OL101:KL47, and the final constructions were confirmed by sequencing as mentioned in Methods. Apart from 2804A>G, no other SNPs were found in the genome of the mutant KP37485-*recC*_{OL101:KL47}. To make it clearer, we have added the description to the text.

Line 163: typo: 1,4407 K. pneumoniae genomes. Currently, there are more than 50 000 genomes of Kp in public repositories.

Reply: Apologize for the typo, and we have corrected it. Indeed, there are more than 50 000 genomes of Kp in public repositories, and in this study we only downloaded the genomes from NCBI RefSeq database as of November 2022. The information has been updated in the text.

Line 164: the allele was found in 763 of 823 ST11 O2v1:KL64 isolates : that is interesting , as it suggests that recC mutation was acquired after the capsule switch event to K64. Could this event also play a role in recombination proficiency, on top of the recC mutation itself? Not sure you have the right controls here. Please indicate the position of the mutation (and of the reference strain 37485) in your tree in Figure 2.

Reply: Thank you for this interesting question. Currently, most of studies suggest that capsule switch is mediated by recombination in *K. pneumoniae*, while it is unclear whether such an event could promote recombination. Based on the recombination analysis results, we suspect that several recombination regions may be associated with capsule switch, however, it could be very difficult to validate it here. The position of the mutation and of the reference strain 37485 has been indicated in the tree in Figure 2.

Please also state clearly if the mutation was never observed outside of the K64 ST11 genomes.

Reply: The sentence has been refined as "...,, and the allele was exclusively found in 763 of 823 ST11 O2v1:KL64 isolates."

Line 175: the statistical tests suppose an independence of observations; here the higher load of MGE is likely to derive from ancestral relationships and vertical descent (after a single acquisition or loss) within the K64 branch, so I would not consider these statistics as being meaningful. The difference in MGEs could simply result from a few acquisition events at the root of the ST11-K64 branch of just later. It would be more meaningful to map the acquisition or loss of the elements along the phylogeny and discuss the relative number of such single events.

Reply: Thank you for the great suggestions. The ancestral state of each MGEs was reconstructed with maximum likelihood using the fastAnc function in the R package phytools v.0.4-98. The results show that the expansions of plasmids and ISs were mediated by a single event in most isolates of ST11-K64, supporting the vertical model. Multiple acquisition and loss events were detected that caused the differences in prophages, suggesting a more complex pattern of horizontal transfer (Figure 4). The results and Figure 4 have been updated in the text, and the original Figure has been changed to Supplementary Figure 5.

Lines 186-188: same comment as just above; please identify unique acquisition events along the phylogeny. It would also provide a view whether the MGE acquisition is still ongoing, or rather ancestral in the K64 branch.

Reply: Thank you for the great suggestions. The ancestral state of each MGE elements was reconstructed with maximum likelihood using the fastAnc function in the R package phytools v.0.4-98. The results show that, as with plasmids and ISs, the expansion of Siphoviridae prophage was mainly mediated by a single event in most isolates of ST11-K64 (Figure 4). The results and Figure 4 have been updated in the text.

Line 192: "potential pathogenesis", as this is genomic observation, rather than phenotypic
Reply: It has been updated according to the reviewer's suggestion.

Line 209: suggesting that the expansion of O2v1:KL64 was driven by the prevalence of the hvKP population: as many inferences on selective value made elsewhere, it is very debatable whether such logical inference is correct; the HV genes could be hitchhiking with other selective markers rather than being the driver; TMP/STX resistance for example is also higher in this branch (Figure 5), a likely driver too.

Reply: We fully understand the reviewer's concern. Here we are trying to infer that the prevalence of O2v1:KL64 is associated with an increase in the size of the hvKP population, based on the observation that "the proportion of O2v1:KL64-hvKP among O2v1:KL64 and CRKP-ST11 dramatically increased from 0% in 2014 to 85.9% (158/184) and 69.9% (158/226) in 2019", but not to infer that the HV genes are the drivers of the expansion of O2v1:KL64. To avoid ambiguity, we have revised the sentence as "....., suggesting that the expansion of O2v1:KL64 was associated with an increase in the size of the hvKP population."

Lines 200 and following, and Figure 2: is it possible to separate rmpA2 and rmpADC in the analysis and results reporting? It is unclear from Fig 2 if it is rmpA2, or rather rmpADC, that is acquired/lost multiple time in the K64 branch.

Reply: Thank you for the comments. The text and figure have been revised as suggested.

Lines 210 and next: how do your plasmid types related to KpVP-1 and KpVP-2 described by Lam et al.?

Reply: According to the definition of Lam et al, the plasmids (encoding *iuc1*) can be typed as KpVP-1.

Figure 5A: please specify which three isolates were used in each of the four columns, and how the 12 triplicates variation was, as this would reinforce your finding.

Reply: Thank you for the comments. Isolates have been listed for the four groups in the legend, and the figure has been updated to show the 12 triplicates variation with error bars as suggested.

Line 240: typo for pVLPK (should be pLVPK)

Reply: We apologize for the typo, and it has been corrected in the text.

Line 247: implying that rmpA/A2-positive O2v1:KL64 might have been selected by certain heavy metals: again, this hypothesis is very risky due to physical genetic association of these genes with others

Reply: Thank you for the comments. The paragraph has been removed.

Line 249: Isolates with sil showed a higher resistance to Ag⁺ (MIC = 0.06 mM) than those without sil (MIC = 0.04 mM): this looks like a very small difference; how to assess its significance? Lines 240-255 could probably be taken out of the work without much harm.

Reply: Thank you for the comments. The paragraph has been removed.

Line 268: again you make an explanatory hypothesis on the emergence, which is made prudently indeed, but might be simply wrong. What is the use of these antimicrobials in China/the studied hospitals? I would suggest to remove these hypotheses from the Results section and to discuss the relative selective advantages provided by all the unique genomic features of K64 branch jointly in the Discussion

Reply: Thank you for the suggestions. These hypotheses have been removed from the Results section, and the Discussion has been updated as suggested.

regarding *oxqAB*, do you have long-read sequencing data supporting their absence/presence and what genetic mechanism (insertion, deletion, or rather, assembly artefact) is involved?

Reply: We have long-read sequencing data for 67 genomes, and 44 of them were confirmed without the *oxqAB* genes. This is consistent with the analysis results using the short-read assemblies. According to the synteny analysis, the genetic mechanism involved in the loss of the *oxqAB* genes seems to be very complicated, and we suspect that ISs and/or recombination would have been involved. The information has been updated in Discussion as "Of note, *oxqAB* have widely been considered to be core genes of *K. pneumoniae*, but our results showed that they were lost in most of the O2v1:KL64 isolates, possibly by IS-mediated and/or recombination mechanisms.", and a supplementary figure (Fig. S15) has been added.

Line 288: where it likely conferred selective advantages: same remark as above

Reply: Thank you for the nice suggestions. Such hypotheses have been removed from the Results section, and the Discussion has been updated as suggested.

Regarding the structure of the Results section, I wonder if the different observations (HV, ARGs, MGEs etc) should not be placed before the *recC* observation/experimental work, which might provide some explanation to the previous observations

Reply: Thank you for the nice suggestions. Indeed, the *recC* observation/experimental work could provide some explanation for the previous observations, but we have no direct evidence to support all the observations except for the recombination proficiency. Therefore, to avoid any misunderstandings, we simply leave this section here and have discussed the suggestions in Discussion.

Line 291-292: Given that the IncFIB-type virulence plasmids and IncFII-type MDR plasmids confer evolutionary fitness to O2v1:KL64: Here you take for granted, without justification, what you previously put forward as simple hypotheses.

Reply: Thank you for the comments. The sentence has been revised as "Given that different genotypes were conferred by the genetic diversity of the IncFIB-type virulence plasmids and IncFII-type MDR plasmids in O2v1:KL64 described above,.....".

Lines 290-301: given the uncertainty in detecting individual genes in short-read assemblies I would be wary of defining so many genotypes precisely. This entire paragraph could be replaced by a discussion section on the existence of 'higher-risk' subtypes carrying a more

complete set of genes of concern. Determining what is selective among these is an entirely different matter.

Reply: Thank you for the comments. The genes detected in the short-read assemblies here belong to ARGs and VFs, and as far as we know, such an analysis strategy has been widely accepted and validated in numerous studies with large-scale genomes (e.g. PMID: 33349664, PMID: 34850838, PMID: 30006589, and PMID: 31358985), even as a routine method for typing, such as cgMLST (e.g. PMID: 25341126). Therefore, detecting individual genes in the short read assemblies should not be a problem.

It is indeed a good suggestion for the “higher-risk” subtypes, however, to avoid the abundant descriptions for the analysis data of this section in Discussion, we will keep the section here. We have revised the section title to “Detection of “high-risk” genotypes in O2v1:KL64”, and the associated content has been updated accordingly.

Lines 303-323: how was the transmission defined, is it based on existence of individual isolates from same/different hospitals within individual genetic clusters?

This entire paragraph should be placed much closer to the beginning of the Results section, as it would enable to define transmission groups/clades that you could use later when describing the gene content of specific tree branches.

Reply: Transmission is defined by pairwise genetic distance (i.e. core-genome SNPs) but not genetic clusters. If the pairwise genetic distance of the most closely related isolate pair is within the threshold, then a transmission event is defined for that pair. Intra-/inter-hospital transmission is classified according to the source (i.e. the same/different hospital) of the isolate pair.

We fully agree with the concern of the reviewer, however, we are unable to move this section before those of the MGEs analysis. This is because the O2v1:KL64-hvKP, which has shown an enhanced inter-hospital transmission, was defined by the results of the MGE analysis. It therefore means that this section must follow the MGEs analysis.

Lines 324-347 should equally be placed much earlier, at the beginning of the Results section, as it would lower the repetitions with the description of the Chinese-only dataset and put it into its global context; the temporal evolution and timing of ancestral nodes is very interesting to have in mind when reading the rest of the Results.

Reply: Thank you for the suggestions. However, we are not sure about “at the beginning of the Results section”. Do you mean that we put this section at the top of the results (before the section “Population structure of CRKP bloodstream (CRKP-BS) isolates in China”)? Could you please clarify it? We could then revise it accordingly.

The fact that another sub-branch independently acquired O2/K64 in Brazil (but elsewhere too it seems from figure S11) also suggests an advantage of this O/K combination. This seems the hypothesis that you put forward at the beginning of the Discussion; but then it is unclear how the other observations you made contribute to the emergence/selective advantage.

Reply: It is indeed an interesting point. The epidemiology of CRKP is known to be highly

geographically dependent. For example, ST258 is prevalent in many countries, but has not been able to spread in China for unknown reasons. Therefore, we should be cautious about using the results of this study to try to explain the epidemiological status in other countries. In addition, although O2/K64 appears to be prevalent in Brazil, it emerged much earlier (2006; Table S9) than in China according to public data. We lack sufficient epidemiological data to assess the potential advantage that this O/K combination might confer in Brazil.

Line 399: pointing to the role of environmental selection in the prevalence of O2v1:KL64-hvKP. Here again, you suppose that all genes provide a selective advantage, although a single driver might be sufficient for each emerging sublineage. In this precise case, environmental selective pressure would suppose transmission between humans and the environment, but you reasoned about mostly hospital transmission in the previous sections.

Reply: Thank you for the comments. The discussion has been revised as suggested by the reviewer.

Lines 429-442: the discussion on transmission seems to be too disconnected from the rest. If a selective advantage was acquired through MGEs, it means the genotype will spread better within hospitals, and then logically more between hospitals too. I don't think you can/need to distinguish advantages that enhance between- rather than within- hospital transmission. What about possible changing patterns of inter-hospital patient transfers that may have played a role?

Reply: We fully agree with the reviewer. This part has been revised as suggested.

Line 445: The epidemic success of O2v1:KL64 is promoted by the emergence and dissemination of a hypervirulence subpopulation : again, this is a hypothesis unduly presented as evidence/conclusion. There is a need to revise entirely the manuscript regarding the selective inferences of genomic (and to a lesser extent, even phenotypic, as they are from in-vitro experiments) observations.

Reply: Thank you for the comments. The selective inferences of genomic observations have been revised throughout the manuscript.

In the Discussion section, it would be critical to discuss the mechanistic role of RecC, and on which type of recombination it acts. I am no expert here, but is recC really acting both on homologous recombination and on MGE mobility, including plasmids and phages, which use their own specific mobility machinery? Unfortunately, the recC finding and its implications in explaining the genomic patterns are not discussed enough, especially considering the title of the manuscript.

Reply: Thank you for the comments. The mechanistic role of RecC has been intensively discussed.

Figure S2: the recombined regions are relative to a reference. Possibly it would be clearer/more in line with the logic of the study, to highlight recombination in the O2/K64 branch, by using a K47 reference.

Reply: The recombination has been reanalyzed using a KL47 reference as suggested,

therefore, the results and the figure has been updated accordingly.

Figure S11: this pattern would suggest that a unique Chinese ST11 branch emerged from global diversity. Is this really the case? I had the inverse notion of emergence in China, followed by global spread. The pattern also suggests multiple independent origins of O2v1/KL64. How was tree constructed, what evolutionary model used, and was recombination purged? Possibly using only representative isolates of the diversity of the Chinese clade (as was done in Figure S12?) would be more appropriate, as the sheer number of genomes in this clade might have brought some phylogenetic artefact.

Reply: Indeed, our results suggest that the Chinese ST11 branched emerged from global diversity, and evolved independently in China. The method of Phylogenetic analysis has been described in Methods, and recombination-free core genome alignment was used to calculate the tree with a GTR model and gamma correction. We repeated the phylogenetic analysis using 150 representative isolates as suggested by the reviewer, and similar results were obtained (see below).

Figure S13: the root-to-tip /time correlation is really not convincing, visually. You should consider providing a demonstration of the temporal signal using a randomization test (i.e., showing that the correlation is stronger with the real dataset than when randomizing sampling dates across the dataset).

Reply: Thank you for the comments. We calculated the root-to-tip /time correlation using the real dataset (i.e. 971 genomes), and found that the correlation is stronger with the real dataset than that with randomizing dataset (0.443 vs 0.3458). The data has been included in the figure.

REVIEWERS' COMMENTS

Reviewer #3 (Remarks to the Author):

The revision version #1 is clearly improved. I still have a number of comments on this version, though.

English language should be thoroughly revised.

Title: please replace 'promotes' by 'associated with'. It is hard to demonstrate the epidemiological effect of a mutation.

Line 34: is 'canonical' useful?

Line 36: "was further bolstered by a selective hypervirulence sublineage": same remark as in the title: please consider association rather than any causation language

Line 42 "driven": same remark

Line 43: please rephrase too; "selective population" has no real meaning and you don't know about the selective driver

Line 311: "selection-associated genes": please remove/rephrase

Line 316: "suggesting that both could be "high-risk" genotypes." High-risk typically means higher infection severity-causing; here you mean something else it seems (more epidemic?). Please remove this notion of 'high-risk' subclones, which to my opinion is not substantiated by data; more successful might be a better terminology

Line 389: "is associated with host selective pressures": please replace by host susceptibility or something similar; there is no evidence for host selective pressure to my opinion (antibodies against OL101 or something similar)

Line 396: (and elsewhere): hypervirulent lineage;

Line 397: only one virulence plasmid, rather?

Line 403: what is "invasion for clinical infection"?

Line 418 : "the highest resistance to tetracycline and sulfamethoxazole-trimethoprim than the others": please revise English language

Line 427: more specifically, what could be the consequences of this loss, knowing oqxAB function?

Line 435-449: thank for you adding this interesting information

Line 453: driving force, not driven force

Line 463: you hypothesize a distinct exposure, but your previous paragraph hypothesized

Line 469 and 474: remove 'high-risk' and replace by successful or similar

Line 476 replace 'produce' by 'incur' or similar

Line 485: MRSA?

Line 503: remove 'hypervirulent'

Line 504: alteration?

Line 641 and Figure 2: it is a bit unfortunate that you used ST258 as an 'outgroup' given that ST258 evolved from within the broader diversity of ST11 and therefore is in fact an 'ingroup'. I guess your rooting is fine because of a large recombination having brought in multiple nucleotide changes to the ST258 genomes, so that they might be placed artefactually as external, but the use of ST258 is theoretically wrong. You might want to use a distinct outgroup or discuss this shortcoming

Fig 4: please place the O2v1/KL64 text on the top left corner of the grey boxes

Fig S2: predicted (typo)

Fig S5: panel Siphoviridae (f) misses the Y-axis legend

Fig S13: MRCA typo

Fig S15: please precise the clusters (O2v1:KL64 etc) of the 4 mentioned isolates, either on the figure or in the legend

REVIEWERS' COMMENTS

Reviewer #3 (Remarks to the Author):

The revision version #1 is clearly improved. I still have a number of comments on this version, though.

English language should be thoroughly revised.

Title: please replace 'promotes' by 'associated with'. It is hard to demonstrate the epidemiological effect of a mutation.

Reply: Thanks for the suggestion. 'promotes' has been replaced by 'associated with'.

Line 34: is 'canonical' useful?

Reply: Thanks for the comment. 'canonical' has been replaced by 'point'.

Line 36: "was further bolstered by a selective hypervirulence sublineage": same remark as in the title: please consider association rather than any causation language

Reply: Thanks for the comment. The sentence has been revised as "The epidemic success of O2v1:KL64 was further associated with a hypervirulent sublineage with.....".

Line 42 "driven": same remark

Reply: Thanks for the comment. "driven" has been replaced by "correlated to".

Line 43: please rephrase too; "selective population" has no real meaning and you don't know about the selective driver

Reply: Thanks for the comment. "selective population" has been replaced by "a subpopulation".

Line 311: "selection-associated genes": please remove/rephrase

Reply: Thanks for the comment. "Selection" has been removed in the text.

Line 316: "suggesting that both could be "high-risk" genotypes." High-risk typically means higher infection severity-causing; here you mean something else it seems (more epidemic?). Please remove this notion of 'high-risk' subclones, which to my opinion is not substantiated by data; more successful might be a better terminology

Reply: Thanks for the suggestion, and "high-risk" has been replaced by "successful" as suggested.

Line 389: "is associated with host selective pressures": please replace by host susceptibility or something similar; there is no evidence for host selective pressure to my opinion (antibodies against OL101 or something similar)

Reply: Thanks for the comment, and "host selective pressures" has been replaced by "host susceptibility".

Line 396: (and elsewhere): hypervirulent lineage;

Reply: Apologize for the typo. We have revised it through the text.

Line 397: only one virulence plasmid, rather?

Reply: Thanks for the comment, and “virulence plasmids“ has been replaced by “a virulence plasmid”.

Line 403: what is “invasion for clinical infection”?

Reply: Apologize for the unclear description. We have replaced “invasion for clinical infection” by “pathogen invasion and subsequent clinical infection”.

Line 418 : “the highest resistance to tetracycline and sulfamethoxazole-trimethoprim than the others”: please revise English language

Reply: Thanks for the comment. The sentence has been revised as “Compared with the other subpopulations, the O2v1:KL64-hvKP displayed the highest resistance to tetracycline and sulfamethoxazole-trimethoprim,”.

Line 427: more specifically, what could be the consequences of this loss, knowing oqxAB function?

Reply: This is indeed an interesting question, however, we don't know much about the consequences of this loss. A brief discussion has been added to the text: “As a multidrug efflux pump, OqxAB confers low to intermediate resistance to several antibiotics (e.g. quinoxalines, quinolones, tigecycline, and nitrofurantoin), detergents and disinfectants. Loss of OqxAB may affect the drug resistance of O2v1:KL64.”.

Line 435-449: thank for you adding this interesting information

Reply: OK.

Line 453: driving force, not driven force

Reply: This has been revised thoroughly.

Line 463: you hypothesize a distinct exposure, but your previous paragraph hypothesized

Reply: This is unclear for us. Do you mean that we hypothesized a distinct exposure in a previous paragraph? We could not find it anywhere, and please show us if this is the case.

Line 469 and 474: remove ‘high-risk’ and replace by successful or similar

Reply: Thanks for the suggestion, and ‘high-risk’ has been replaced by ‘successful’.

Line 476 replace ‘produce’ by ‘incur’ or similar

Reply: Thanks for the suggestion, and ‘incur’ has been replaced by ‘produce’.

Line 485: MRSA?

Reply: Apologize for the typo. ‘MRAS’ has been revised as ‘MRSA’.

Line 503: remove ‘hypervirulent’

Reply: Thanks for the suggestion, and 'hypervirulent' has been removed.

Line 504: alteration?

Reply: Apologize for the typo, and 'alternations' has been revised as 'alterations'.

Line 641 and Figure 2: it is a bit unfortunate that you used ST258 as an 'outgroup' given that ST258 evolved from within the broader diversity of ST11 and therefore is in fact an 'ingroup'. I guess your rooting is fine because of a large recombination having brought in multiple nucleotide changes to the ST258 genomes, so that they might be placed artefactually as external, but the use of ST258 is theoretically wrong. You might want to use a distinct outgroup or discuss this shortcoming

Reply: We agree with the reviewer that ST258 evolved from within the broader diversity of ST11. However, we believe that ST258 is a desirable outgroup for our analysis, particularly at subclonal level. As defined, outgroups are taxa that diverged from all ingroup taxa before they diverged from each other and are the preferred way to determine the root of a phylogenetic tree (Schneider and Cannarozzi, 2009). Additionally, it has been previously proposed that from all possible outgroups of comparable sequencing quality, the closest one is the best choice to determine a rooted tree (Ritland and Clegg 1991; Muse and Weir 1992; Smith 1994) because shorter distances suffer less from statistical error and also the expected number of homoplasies between any ingroup and the outgroup is minimized this way. These arguments support our choice of ST258 as a desirable outgroup.

We also utilized an ST10 isolate as an outgroup, which has a more distant relationship with ST11. The topology of the resulting phylogenetic tree is similar to our current tree (as shown below). As a result, we have decided to keep the figure unchanged.

Reference:

1. Adrian Schneider, Gina M. Cannarozzi, Support Patterns from Different Outgroups Provide a Strong Phylogenetic Signal, *Molecular Biology and Evolution*, Volume 26, Issue 6, June 2009, Pages 1259–1272, <https://doi.org/10.1093/molbev/msp034>
2. Ritland K, Clegg MT, O'Brien SJ. Optimal DNA sequence divergence for testing phylogenetic hypotheses, *Molecular Evolution*, 1991 New York Wiley-Liss (pg. 289-296)
3. Muse SV, Weir BS. Testing for equality of evolutionary rates, *Genetics*, 1992, vol. 132 (pg. 269-276)
4. Smith AB. Rooting molecular trees: problems and strategies, *Biol J Linn Soc.*, 1994, vol. 51 (pg. 279-292)

Fig 4: please place the O2v1/KL64 text on the top left corner of the grey boxes

Reply: The figure has been updated as suggested.

Fig S2: predicted (typo)

Reply: Apologize for the typo, and 'predicated' has been revised as 'predicted'.

Fig S5: panel Siphoviridae (f) misses the Y-axis legend

Reply: The figure has been updated.

Fig S13: MRCA typo

Reply: Apologize for the typo, and 'MRACs' has been revised as 'MRCAs'.

Fig S15: please precise the clusters (O2v1:KL64 etc) of the 4 mentioned isolates, either on the figure or in the legend

Reply: Thanks for the suggestion. The legend has been updated as suggested.